# MPMAvatar: Learning 3D Gaussian Avatars with Accurate and Robust Physics-Based Dynamics

**Changmin Lee**     **Jihyun Lee**     **Tae-Kyun Kim**

KAIST

{lcm914, jyun.lee, kimtaekyun}@kaist.ac.kr

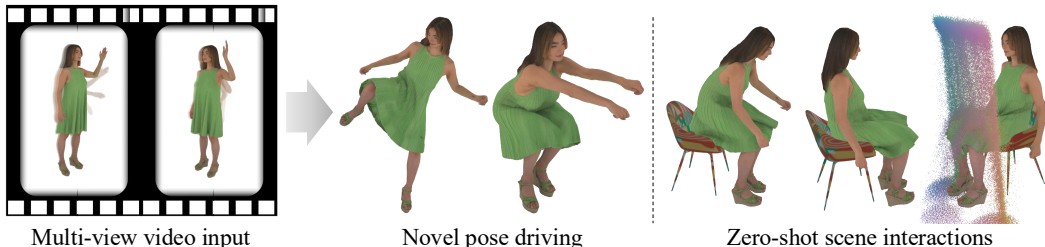

Figure 1. **3D Gaussian avatars with accurate and robust physics-based animation.** We present MPMAvatar, a framework for creating 3D Gaussian avatars from multi-view videos, that support *physically accurate and robust animations*, especially for loose garments. Ours is also zero-shot generalizable to novel scene interactions.

## Abstract

While there has been significant progress in the field of 3D avatar creation from visual observations, modeling physically plausible dynamics of humans with loose garments remains a challenging problem. Although a few existing works address this problem by leveraging physical simulation, they suffer from limited accuracy or robustness to novel animation inputs. In this work, we present MPMAvatar, a framework for creating 3D human avatars from multi-view videos that supports highly realistic, robust animation, as well as photorealistic rendering from free viewpoints. For accurate and robust dynamics modeling, our key idea is to use a Material Point Method-based simulator, which we carefully tailor to model garments with complex deformations and contact with the underlying body by incorporating an anisotropic constitutive model and a novel collision handling algorithm. We combine this dynamics modeling scheme with our canonical avatar that can be rendered using 3D Gaussian Splatting with quasi-shadowing, enabling high-fidelity rendering for physically realistic animations. In our experiments, we demonstrate that MPMAvatar significantly outperforms the existing state-of-the-art physics-based avatar in terms of (1) dynamics modeling accuracy, (2) rendering accuracy, and (3) robustness and efficiency. Additionally, we present a novel application in which our avatar generalizes to unseen interactions in a zero-shot manner—which was not achievable with previous learning-based methods due to their limited simulation generalizability. Our project page is at: `https://KAISTChangmin.github.io/MPMAvatar/`.

## 1 Introduction

Creating 3D human avatars has been an important research problem due to their broad range of applications, including virtual and augmented reality, computer games, and content creation. The key goals of this field include modeling and learning avatars that support (1) photorealistic rendering from free viewpoints and (2) realistic animation driven by sparse motion inputs (e.g., SMPL-X [51]

39th Conference on Neural Information Processing Systems (NeurIPS 2025).

parameters). On the rendering side, recent approaches based on 3D Gaussian Splatting [22] have shown remarkable progress in enabling high-fidelity avatar rendering from free viewpoints. However, realistically animating avatars under novel motion inputs remains a challenging problem, especially for loose garments. Existing methods typically model these garment motions using piecewise linear transformations [1, 3, 6, 48, 61, 72, 47, 73] or pose-dependent geometric correctives [5, 35, 38, 39, 65], but they are known to be limited in accurately capturing complex deformations and tend to overfit to motions observed during training, as discussed in [78].

To address these limitations, incorporating physics-based prior via *physically simulating* the avatar dynamics can be an effective approach to enhance the realism and generalizability of animation. There are only a few works towards this direction; Xiang *et al.* [69] proposes an avatar simulated using X-PBD [41], but their method relies on a time-consuming manual parameter search to approximate reasonable cloth behavior. PhysAvatar [78] is the most recent physics-based avatar that adopts C-IPC [31] for garment simulation. However, the simulator fails when the animation inputs, SMPL-X [51] meshes, have a small degree of self-penetration, which can occur as they are practically obtained via *estimation* (e.g,. fitting the parametric model to input observations). This causes C-IPC to fail to resolve the collision in the Continuous Collision Detection (CCD) stage [31], thus the driving body meshes are *manually adjusted* to avoid simulation failures in this method. In terms of appearance, PhysAvatar relies on mesh-based rendering, which limits its ability to capture fine-grained appearance details.

In this paper, we propose MPMAvatar, a framework for creating 3D human avatars from multi-view videos that enables (1) physically accurate and robust animation especially for loose garments, as well as (2) high-fidelity rendering based on 3D Gaussian Splatting [22]. For garment dynamics modeling, our key idea is to use a Material Point Method [17] (MPM)-based simulator, which is capable of simulating objects under complex contacts without failure cases via feedforward velocity projection. However, directly adopting the existing MPM simulator [17], mainly used for general object dynamics modeling, introduces two challenges for effective garment dynamics modeling for our avatar.

First, garments typically exhibit a codimensional manifold structure, and their physical properties vary significantly depending on the direction (e.g., in-manifold vs. normal directions). Second, the existing collision handling algorithm [17] of MPM is mainly designed for *colliders* that are analytically represented using level sets (e.g., simple geometries such as spheres), which is not applicable in our scenario where the collider is SMPL-X [51] body mesh underlying the garments. To address these issues, we tailor our MPM simulator by (1) adopting an anisotropic constitutive model [16] to better model the manifold-dependent dynamics of garments, and (2) by introducing a novel collision handling algorithm that can handle more general colliders represented as meshes. We combine this dynamics modeling scheme with our canonical avatar that can be rendered using 3D Gaussian Splatting with quasi-shadowing, enabling high-fidelity rendering for physically realistic animations.

In our experiments, we demonstrate that MPMAvatar outperforms the current state-of-the-art physics-based avatar [78] in terms of (1) dynamics modeling accuracy, (2) rendering accuracy, and (3) simulation robustness and efficiency. Additionally, we present a novel application in which our avatar generalizes to novel scene interactions in a zero-shot manner – demonstrating the generalizability of our physics simulation-based dynamics modeling. Our code will be also publicly available to allow full reproducibility.

Overall, our contributions can be summarized as follows:

- We present MPMAvatar, a novel framework for creating 3D clothed human avatars from multi-view videos. Our avatar supports physically realistic and robust animations, especially for loose garments, as well as high-quality rendering.
- We present an MPM [17]-based simulation method carefully tailored for effective garment dynamics modeling, based on an anisotropic constitutive model [16] and mesh-based collider handling.
- We empirically demonstrate that MPMAvatar achieves superior performance than the existing SOTA physics-based avatar (PhysAvatar [78]) in terms of (1) dynamics modeling accuracy, (2) rendering accuracy, and also (3) simulation robustness and efficiency.
- We show that our physics-based simulation method is *zero-shot generalizable* to interactions with an unseen external object. To the best of our knowledge, MPMAvatar is the first to empirically demonstrate this ability.

## 2    Related Work

**Physics-based simulation methods.** Physics-based simulation methods [46, 41, 21, 30, 31, 17] simulate object dynamics by solving the governing differential equations (e.g., conservation of momentum). Position-Based Dynamics (PBD) [46, 41] directly manipulates the positions of particles to satisfy physical constraints. While it is fast and stable, it is limited in accurately modeling complex material behaviors due to the lack of explicit physical modeling (e.g., plasticity). Variational integrators [21], such as IPC [30] and its cloth-specific extension C-IPC [31], formulates simulation as the iterative minimization of a total energy potential, which is possibly augmented with barrier terms to robustly handle contact and friction. While C-IPC is effective for simulating cloth and was thus adopted in the current SOTA physics-based avatar [78], it fails to resolve collisions for *noisy* colliders during Continuous Collision Detection (CCD), as discussed in Sec. 1. Some recent works [37, 33] explore differentiable cloth simulation for inverse problems. However, they primarily operate in simplified settings without complex colliders, making them less applicable to collision-heavy scenarios like clothed human avatars. Material Point Method (MPM) [17] simulates continuum materials by hybridly representing them with Lagrangian particles and an Eulerian grid, and is known for robust handling of large deformations and self-collisions. Due to these advantages crucial for simulation stability and accuracy, our work adopts MPM, while further tailoring its simulation scheme for more effective garment dynamics modeling for our avatar. In a later section (Sec. 5), we demonstrate that our simulation method outperforms C-IPC used in the SOTA physics-based avatar [78] in both dynamics modeling accuracy and robustness.

**Learning-based simulation methods.** Some existing methods [8, 7] use neural network-based simulators trained on large datasets to *implicitly* encode physics. However, they have limited generalizability beyond the training dynamics and cannot *guarantee* physically plausible deformations, as object dynamics are *predicted* by the network. As discussed in [8], this leads to failure cases when objects move faster or more erratically than in the training data. Thus, directly following the motivation of the recent physics-based avatar [78], we adopt a physics-based simulation. In Sec. 5, we additionally show that our MPM-based simulation method is *zero-shot generalizable* to novel scene interactions — which is not achievable through learning-based simulation.

**Dynamic avatar reconstruction from visual inputs.** Learning 3D human avatars from visual inputs has been an active area of research in computer vision and graphics. Earlier methods rely on (1) *a mesh representation* [51, 9, 11, 56, 71, 10, 24, 29, 23, 62], which is computationally efficient but struggles to capture details for rendering, or (2) *an implicit representation* [13, 52, 68, 77, 53, 18, 27, 28], which captures high-frequency geometry and appearance details but complicates dynamics modeling due to the absence of explicit geometry. Most recent avatar reconstruction methods [20, 36, 45, 49, 79] use 3D Gaussian Splats (3DGS) [22], which address these limitations by enabling high-fidelity rendering while using explicit representation, facilitating more effective dynamics modeling. For modeling avatar dynamics, existing methods often represent garment motions using piecewise linear transformations [1, 3, 6, 48, 61, 72, 47, 73] or pose-dependent geometric correctives [5, 35, 38, 39, 65]. However, these approaches are limited in their ability to capture complex deformations and tend to overfit to the training motion data, as discussed in [78]. To enable more realistic and physically accurate animations, a few works [58, 69, 78] have integrated simulators such as XPBD [41, 58], C-IPC [31, 70]. However, they require manual parameter search, or manual adjustment for body mesh colliders to avoid simulation failures, as discussed in Sec. 1. A related line of work, such as DiffAvatar [34], focuses on generating physically plausible garments from static 3D scans using differentiable simulation. However, their formulation omits appearance modeling and is tailored for scan-based asset preparation, rather than dynamic avatar reconstruction from visual observations. Note that some existing works on garment-only modeling (not the clothed avatars as in our work) adopt a neural network-based simulator [8, 7, 55], but they do not *guarantee* physically based behavior and are known to have limited generalizability compared to physics-based simulators [78].

## 3    Preliminaries: Material Point Method (MPM)

Material Point Method (MPM) [17] models an object as a continuum, enabling the simulation of diverse materials including solids, liquids, and gases. MPM advances the simulation by representing the continuum using both Lagrangian particles and an Eulerian grid and solving two governing equations: (a) conservation of mass and (b) conservation of momentum:

$$(a) \quad \frac{D\rho}{Dt} + \rho \nabla \cdot \mathbf{v} = 0, \qquad (b) \quad \rho \frac{D\mathbf{v}}{Dt} = \nabla \cdot \boldsymbol{\sigma} + \rho \, \mathbf{g}, \tag{1}$$

where $\rho$ and $\mathbf{v}$ are density and velocity, respectively. $\frac{D}{Dt}$ denotes the material derivative, $\boldsymbol{\sigma}$ is the cauchy stress tensor, and $\mathbf{g}$ is the gravitational acceleration. During simulation, physical quantities such as mass and momentum are two-way transferred between the particles and the grid. Here, mass conservation is straightforwardly achieved due to the invariant mass carried by Lagrangian particles, while momentum conservation is performed on the Eulerian grid to efficiently approximate the spatial derivatives. When solving Eq. 1b for updating momentum on the grid, computing the cauchy stress tensor $\boldsymbol{\sigma}$ is the key in capturing the material behavior. More specifically, it is defined as $\boldsymbol{\sigma} = \frac{1}{\det(\mathbf{F})} \frac{\partial \psi}{\partial \mathbf{F}} \mathbf{F}^T$, where $\mathbf{F}$ is the deformation gradient that linearly approximates local deformations, and the strain-energy density function $\psi$, which depends on $\mathbf{F}$, quantifies the energy stored through the deformation. For defining $\psi$, various *constitutive models* [59, 25, 60] have been developed to define $\psi$, such that it can effectively model various material behaviors (e.g. for jelly, snow, sand, and fluids).

## 4    MPMAvatar: Photorealistic Avatars with Physics-Based Dynamics

In this section, we present MPMAvatar, a framework that learns 3D human avatars from multi-view videos that support (1) physically accurate and robust animation and (2) high-quality rendering. In the following sections, we first describe our avatar representation (Sec. 4.1). We then present our physics-based approach to modeling avatar dynamics, which is particularly effective for realistically animating loose garments (Sec. 4.2). Finally, we explain how the proposed physically-based dynamic avatar can be learned from multi-view video inputs (Sec. 4.3).

### 4.1    Avatar Representation

To enable both physically realistic animation and high-fidelity rendering, we use a hybrid representation that combines (1) a mesh with physical parameters to enable physically based animation, and (2) 3D Gaussian Splats [22] for high-quality rendering. Formally, we represent the canonical geometry of an avatar with a 3D triangular mesh $\mathcal{M}_1 = (\mathbf{V}_1, \mathbf{F})$, where $\mathbf{V}_1$ and $\mathbf{F}$ are mesh vertices and faces, respectively. [1] To model the physics-based dynamics of the geometry, the avatar is also represented with physical parameters $\mathcal{P} = (E, \nu, \gamma, \kappa, \rho, \alpha)$. It consists of material parameters used for traditional Material Point Method (MPM) [17] simulation (Young's modulus $E$ and Possion's ratio $\nu$), additional material parameters for anistropic dynamics modeling (shear stiffness $\gamma$ and normal stiffness $\kappa$, which will be introduced in Sec. 4.2.1), density $\rho$, and the rest geometry parameter $\alpha$.

To enable high-fidelity rendering, we represent the avatar appearance with 3D Gaussian Splats [22] $\mathcal{G} = \{\mathbf{g}_i\}_{i=1\cdots N_G}$. Each Gaussian Splat $\mathbf{g}_i$ is parameterized by a translation vector $\mathbf{t}_i \in \mathbb{R}^3$, a quaternion $\mathbf{q}_i \in \mathbb{R}^4$, a scale vector $\mathbf{s}_i \in \mathbb{R}^3$, an opacity $o_i \in \mathbb{R}$, and color $\mathbf{c}_i$ represented by spherical harmonics [22]. Note that our Gaussian Splats are *attached* to the canonical avatar mesh $\mathcal{M}_1$, where each $\mathbf{g}_i$ is associated with a face of $\mathcal{M}_1$. More specifically, following [54], we define all spatial parameters of $\mathbf{g}_i$ in the local coordinate system with respect to its associated mesh face, allowing our Gaussian Splats to naturally deform according to the underlying mesh deformations.

### 4.2    Physics-Based Dynamics Modeling

We now explain how we model the dynamics of our avatar discussed in Sec. 4.1 – to achieve highly realistic and physically grounded animations. Following the existing physics-based avatar [78], we animate the body (non-garment regions) of the avatar using Linear Blend Skinning [15], while animating its garments driven by the underlying body motions (represented with SMPL-X [51] meshes) via physical simulation. For the simulation, we adopt Material Point Method (MPM) [17] due to its effectiveness in modeling large deformations and robustly handling collisions (Sec. 2). While MPM is actively adopted in recent 3D scene simulation methods [4, 76, 70], it is mainly used for modeling the dynamics of general objects (e.g., flower pots, elastic torus).

---

[1] While the deformation gradient is also denoted by $\mathbf{F}$, we allow a slight abuse of notation to remain consistent with notation conventions used in related work.

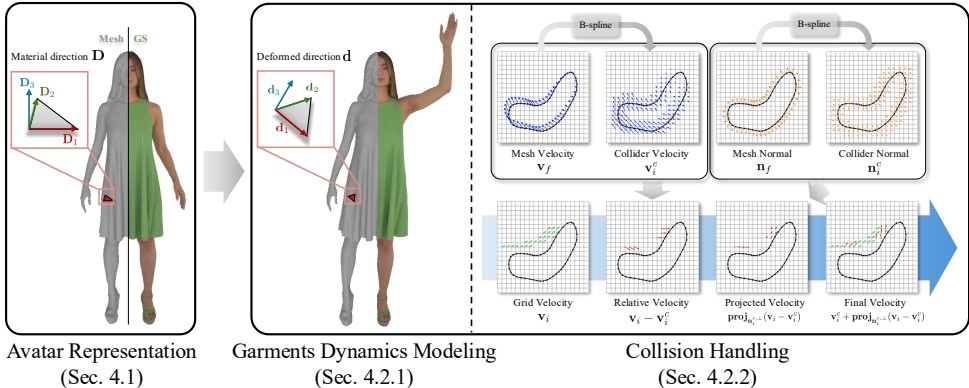

Figure 2. **Overview of our dynamic avatar modeling.** We hybridly represent our canonical avatar with (1) a mesh with physical parameters for geometry and dynamics modeling, and (2) 3D Gaussian Splats [22] for appearance modeling (Sec. 4.1). This avatar can be animated via linear blend skinning for non-garment regions and physical simulation for garment regions (Sec. 4.2.1) with our novel collision handling algorithm (Sec. 4.2.2). *Visualization key.* Blue arrows indicate body grid velocities, green arrows denote garment grid velocities, and red arrows show colliding grid regions where velocity projection is applied.

In this work, we carefully tailor the simulator [17] to achieve more effective modeling of *garment dynamics* in our avatar. In particular, we (1) adopt the anisotropic constitutive model [16] to better model the manifold-dependent dynamics of garments, and (2) introduce a collision handling algorithm designed to effectively resolve garment-body collisions. In what follows, we further elaborate on these two modifications.

### 4.2.1 Anisotropic Dynamics Modeling for Garments

Garments typically exhibit a codimensional manifold structure, and their physical properties vary depending on it. For example, garments can easily stretch along in-manifold directions, but not along the normal directions. To accurately model this behavior, we adopt the *anisotropic* constitutive model proposed by Jiang *et al.* [16] underlying our MPM simulator. In particular, they propose to model the strain energy for anisotropic material depending on each Lagrangian particle's material directions, which are approximated using Lagrangian mesh. More specifically, the deformation gradient $\mathbf{F}$ at each particle is computed as $\mathbf{F} = \mathbf{d}\mathbf{D}^{-1}$, where $\mathbf{D} = [\mathbf{D}_1, \mathbf{D}_2, \mathbf{D}_3] \in \mathbb{R}^{3\times3}$ is the original material direction and $\mathbf{d} = [\mathbf{d}_1, \mathbf{d}_2, \mathbf{d}_3] \in \mathbb{R}^{3\times3}$ is the deformed material direction (see Fig. 2). Since the strain-energy density function $\psi$ must be invariant under rotations, they apply QR-decomposition $\mathbf{F} = \mathbf{Q}\mathbf{R}$, and $\psi$ is reparameterized as $\psi(\mathbf{F}) = \hat{\psi}(\mathbf{R})$, such that:

$$\hat{\psi}(\mathbf{R}|E, \nu, \kappa, \gamma) = \hat{\psi}_{\text{normal}}(\mathbf{R}_{33}|\kappa) + \hat{\psi}_{\text{shear}}(\mathbf{R}_{13}, \mathbf{R}_{23}|\gamma) + \hat{\psi}_{\text{in-plane}}(\mathbf{R}_{11}, \mathbf{R}_{12}, \mathbf{R}_{22}|E, \nu),$$

where $\hat{\psi}_{\text{normal}}$, $\hat{\psi}_{\text{shear}}$, and $\hat{\psi}_{\text{in-plane}}$ are functions for penalizing normal deformation, shearing, and in-plane deformation, respectively (refer to [16] and Appendix C for details). Note that this constitutive model [16] requires a Lagrangian mesh representing the codimensional object to track the material directions, which had motivated our avatar representation (Sec. 4.1) based on a hybrid mesh and Gaussian Splats. As the official implementation of the MPM solver for this constitutive model is not publicly available, we re-implemented it using PyTorch [50] and Warp [40], and plan to release our code to facilitate future research.

### 4.2.2 Collision Handling

We additionally introduce a collision handling algorithm designed to effectively resolve our garment-body collisions.

**Collision handling of MPM [17].** The existing collision handling algorithm of MPM is designed to be effective for an external object (i.e., a collider) *represented as a dynamic level set*. In a nutshell, MPM resolves collisions by projecting the grid velocity of the *colliding region* of the object onto the

collider's tangent space, preventing penetration while modeling tangential motion. Technically, this requires evaluating the velocity and normal of the collider at the Eulerian grids *nearby* the colliding regions, which are not strictly on the collider surface. MPM originally enables these evaluations by considering colliders whose geometry and velocity fields are analytically defined over all points $\mathbf{x} \in \mathbb{R}^3$ in the ambient 3D space, e.g., a sphere geometry $\phi(\mathbf{x}) = \|\mathbf{x} - \mathbf{c}\| - r$, where $\mathbf{c}$ and $r$ denote center and radius, respectively. However, our collider is a SMPL-X [51] mesh representing the body underlying the garment, thus it is not trivial to directly adopt this algorithm.

**Our algorithm.** To address the above limitation, we introduce a simple yet effective collision handling algorithm for MPM to support colliders represented as meshes. Note that, in this case, normal or velocity is defined only on the collider surface, not at all Euclidean grid nodes. To address this, our idea is to transfer these quantities to nearby grid nodes using B-spline weights, directly analogical to *particle-to-grid transfer* in MPM simulation. In Fig. 2, we outline our overall collision handling procedure consisting of two stages: (1) mesh–to–grid transfer (upper row) and (2) relative velocity projection (lower row). In the mesh-to-grid transfer stage, we transfer each collider face's velocity $\mathbf{v}_f$ and normal $\mathbf{n}_f$ to nearby grid nodes using B-Spline weights, producing the extended collider velocity $\mathbf{v}_i^c$ and normal $\mathbf{n}_i^c$ at each grid node $i$. In the relative velocity projection stage, we first transform the grid velocity into the collider mesh's reference frame by subtracting the collider velocity $\mathbf{v}_i^c$. If the relative velocity points inward, we project out its normal component. Finally, we transform the corrected velocities back into the world frame. It is worth noting that the complexity of the existing collision handling algorithm of MPM is $O(N_{\text{grid}}^3)$, where $N_{\text{grid}}$ is the grid resolution, as it requires evaluating the level set function at all grid nodes for collision check. In contrast, the complexity of our collision handling algorithm is $O(N_{\text{f}})$, where $N_{\text{f}}$ is the number of the collider mesh faces. As it is usually $N_{\text{f}} \ll N_{\text{grid}}^3$ (note that $N_{\text{f}} \approx 20\text{K}$ and $N_{\text{grid}}^3 \approx 8\text{M}$ in our case), our approach based on a mesh-based collider is extremely more efficient, as well as reflecting a more practical scenario.

**Summary.** Using our tailored MPM-based simulator, we can effectively model the anisotropic dynamics of garments under complex collisions with the underlying body meshes. Later in the experiments (Sec. 5), we show that our simulation scheme leads to SOTA dynamics accuracy.

### 4.3 Learning from Multi-View Videos

We now explain how our avatar outlined in the previous sections can be learned from multi-view video inputs. As a preprocessing step, we first perform 3D mesh tracking on the input frames to capture dense temporal geometry correspondences, which are used to supervise the subsequent learning stages. In particular, we use the mesh tracking algorithm of the existing physics-based avatar work [78], which assumes that a mesh at the first frame $\mathcal{M}_1 = (\mathbf{V}_1, \mathbf{F})$ is given (e.g., from an off-the-shelf static scene reconstruction method), and optimizes the deformed meshes at the subsequent frames $(\mathcal{M}_i)_{i=2\cdots T}$, where $\mathcal{M}_i = (\mathbf{V}_i, \mathbf{F})$, based on a rendering loss. For more details on the tracking algorithm, we refer the reader to [78]. In the following, we focus on explaining how the physical dynamics (Sec.4.3.1) and appearance (Sec.4.3.2) of our avatar can be learned.

#### 4.3.1 Physical Parameters Learning

We now explain how we learn the physical parameters used to model our garment dynamics. Given the canonical avatar mesh $\mathcal{M}_1$ obtained from the prior mesh tracking stage, the set of physical parameters associated with $\mathcal{M}_1$ is defined as $\mathcal{P} = (E, \nu, \gamma, \kappa, \rho, \alpha)$, as previously discussed in Sec. 4.1. Note that $E, \nu, \gamma$, and $\kappa$ are material parameters used for our Material Point Method (MPM) [17] simulation. Following the recent inverse physics work based on MPM [76], which found that Young's modulus $E$ is the key parameter dominating the dynamic behavior and thus fixed the rest of the parameters, we also fix $\nu, \gamma, \kappa$ to their default values and focus on learning the other parameters—$E, \rho$, and $\alpha$ to mitigate over-parameterization. In the following, we specifically focus on elaborating on our newly introduced parameter $\alpha$, used for rest geometry modeling.

**Rest geometry modeling.** MPM [17] internally computes forces based on the deformation gradient relative to the object's *rest geometry* (i.e., the canonical geometry in the unstressed state, without external forces such as gravity). Note that existing approaches [4, 76] based on MPM typically assume ideal conditions, i.e., having the initial frame correspond to an undeformed rest state, limiting

their applicability in real-world scenarios. In our case, the canonical geometry $\mathcal{M}_1$ is obtained from real-world observations and is therefore already deformed by gravity. To correct this, we additionally introduce a simple parameter $\alpha \in [0, 1]$ to compensate for gravity-induced deformation and learn the unseen rest geometry of the avatar. Formally, for each edge vector $\mathbf{e}$ in $\mathcal{M}_1$, we decompose it into two components: $\mathbf{e}_g$, the projection of $\mathbf{e}$ onto the gravity direction $\mathbf{g}$, and $\mathbf{e}_\perp$, the component orthogonal to $\mathbf{g}$, such that $\mathbf{e} = \mathbf{e}_g + \mathbf{e}_\perp$. Using $\alpha$, we simply model each edge $\mathbf{e}_{\text{rest}}$ in the rest geometry as: $\mathbf{e}_{\text{rest}} = \mathbf{e}_\perp + \alpha\,\mathbf{e}_g$, where $\alpha$ determines the extent to which the stretch in the gravity direction is compensated. $\alpha$ is optimized end-to-end along with other physical parameters via inverse physics.

**Learning physical parameters.** We aim to learn $E$, $\rho$, and $\alpha$ such that the simulated mesh dynamics closely model the real-world garment motions. Given the canonical mesh at the first frame $\mathcal{M}_1$, we simulate its future states using our MPM-based simulator (Sec. 4.2) and the physical parameters $\mathcal{P}$, resulting in simulated meshes $(\hat{\mathcal{M}}_i)_{i=2,\dots,T}$ over frames $[2, T]$. We then optimize the parameters $E$, $\rho$, and $\alpha$ by minimizing the vertex-to-vertex L2 loss between the simulated meshes and the tracked meshes $(\mathcal{M}_i)_{i=2,\dots,T}$ capturing geometric dynamics from the input video. Following [78], we perform this optimization using a finite-difference approach. For more implementation details, we kindly refer the reader to Appendix B.1.

### 4.3.2 Appearance Learning

For learning the appearance of our avatar, we optimize the parameters of 3D Gaussian Splats $\mathcal{G}$ defined on the canonical geometry $\mathcal{M}_1$ at $t = 1$. As discussed in Sec. 4.1, the spatial parameters of $\mathbf{g}_i$ are defined in the local coordinate system with respect to its parent mesh face, allowing it to naturally deform according to the underlying mesh deformations. Using this, we first deform $\mathcal{G}$ to the other frames at $t \in \{2, \dots, T\}$ based on the tracked mesh deformations $(\mathcal{M}_i)_{i=1,\dots,T}$, and render them across all input views and timesteps. We then compute the loss by measuring the photometric discrepancy between the rendered images and the ground-truth images across all training frames and views. We finally optimize the parameters of $\mathcal{G}$ via gradient descent to minimize this loss.

Note that the preprocessing mesh tracking stage, which we adopt from [78], also employs 3D Gaussian Splats as a surrogate representation to incorporate rendering loss for optimizing meshes. However, it learns the Gaussian colors *independently for each frame*[1]. In contrast, we learn $\mathcal{G}$ from all input frames and views to better capture regions that are occluded in some views but visible in others, while still leveraging the previously learned surrogate Gaussian Splats for parameter initialization. Please refer to Appendix B.2 for the details.

**Quasi-shadowing.** When rendering our avatar using $\mathcal{G}$ via 3D Gaussian Splatting, we additionally apply quasi-shadowing to enhance rendering fidelity. Following the prior work [2], we model self-shadowing by leveraging a neural network trained on ambient occlusion features extracted from the mesh in our hybrid avatar representation. Specifically, we modulate the color of each Gaussian Splat $g_i$ using a shading scalar $w_p \in [0, 1]$, predicted by the network, to obtain the final color.

## 5 Experiments

### 5.1 Experimental Setup

For experimental comparisons, we mainly follow the setup used in the state-of-the-art physics-based avatar work (PhysAvatar [78]) to perform fair comparisons.

**Dataset.** We perform our main evaluations on **(1) ActorsHQ [14]**. In particular, we select four subjects used in [78]: two characters in loose dresses and two characters in two-piece outfits. For training each subject, we use 24 frames with large cloth dynamics for physical parameter learning and 200 frames for appearance learning. For testing, we use 200 unseen frames per subject. Whereas the existing work [70] only uses ActorsHQ for evaluation, we additionally include four sequences from **(2) 4D-DRESS [66]** dataset, to perform more extensive comparisons. We use two subjects in tops and skirts and two in tops and tight jeans. For training, we use 11 frames for physical parameter learning and 100 frames for appearance learning, while testing was carried out on 100 unseen frames.

---

[1]Note that the existing work [78] with this mesh tracking method discards the surrogate Gaussian Splats and resorts to mesh-based rendering for its avatar.

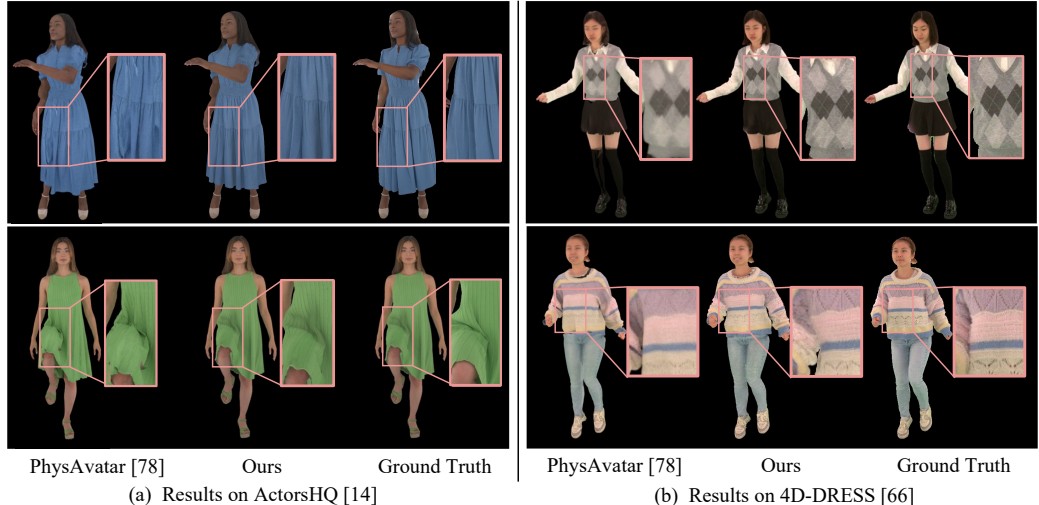

| PhysAvatar [78] | Ours | Ground Truth | PhysAvatar [78] | Ours | Ground Truth |

(a) Results on ActorsHQ [14]  (b) Results on 4D-DRESS [66]

Figure 3. **Qualitative results on test frames in the ActorsHQ [14] and 4D-DRESS [66] datasets.** Our method outperforms PhysAvatar [78] in appearance by rendering sharper, less blurred textures with finer detail and in geometry by recovering folds and wrinkles that more closely match the ground truth.

Table 1. **Quantitative comparisons on ActorsHQ [14] and 4D-DRESS [66] datasets.** Bold indicates the best scores, and underline indicates the second best scores. Our proposed method achieves the best results across all geometry and appearance metrics on both benchmarks.

| | Method | Geometry | | Appearance | | |
| | | CD ($\times 10^3$) ↓ | F-Score ↑ | LPIPS ↓ | PSNR ↑ | SSIM ↑ |
|---|---|---|---|---|---|---|
| | **(a) Results on ActorsHQ [14] dataset.** | | | | | |
| A | ARAH [65] | 1.12 | 86.1 | 0.055 | 28.6 | 0.957 |
| B | TAVA [32] | 0.66 | 92.3 | 0.051 | 29.6 | 0.962 |
| C | GS-Avatar [12] | 0.91 | 89.4 | 0.044 | 30.6 | 0.962 |
| D | PhysAvatar [78] | 0.55 | 92.9 | 0.035 | 30.2 | 0.957 |
| E | **MPMAvatar (Ours)** | **0.42** | **95.7** | **0.033** | **32.0** | **0.963** |
| F | — Anisotropy | 6.24 | 90.3 | 0.039 | 28.7 | 0.957 |
| G | — Physics | 0.69 | 92.9 | 0.039 | 31.0 | 0.962 |
| H | — Shadow | - | - | **0.033** | 31.8 | **0.963** |
| | **(b) Results on 4D-DRESS [66] dataset.** | | | | | |
| A | PhysAvatar [78] | 0.37 | 96.6 | 0.022 | 33.2 | 0.976 |
| B | **MPMAvatar (Ours)** | **0.33** | **97.2** | **0.018** | **34.1** | **0.977** |

**Baselines.** We use the same baselines as in [78], which are four open-sourced avatar reconstruction methods: ARAH [65], TAVA [32], GS-Avatar [12], and PhysAvatar [78]. Here, PhysAvatar is the most related baseline to ours, as it is the state-of-the-art work on physics-based avatar. As we already discussed in Sec. 1, PhysAvatar's simulator fails when driving body mesh colliders have self-penetrations. While the original work manually adjusted the body meshes to avoid simulation failures, these meshes are not publicly available after requests. Therefore, we minimally excluded the faces of the body mesh collider during collision check to prevent their simulation failure.

**Evaluation Metrics.** To assess our dynamics modeling accuracy, we compute Chamfer Distance (CD) [44] and F-Score [64] between the simulated and the ground truth meshes. For F-Score, we set the threshold $\tau$ to 0.001. For evaluating our rendering accuracy, we measure Learned Perceptual Image Patch Similarity (LPIPS) [75], Peak Signal-to-Noise Ratio (PSNR), and Structural Similarity Index Measure (SSIM) between the rendered and the ground truth images.

## 5.2 Experimental Comparisons

**Simulation and rendering accuracy.** Tab. 1(a) (Rows A-E) and Tab. 1(b) show our main comparison results on the ActorsHQ [14] and 4D-DRESS [66] datasets, respectively. Ours achieves the best results across all *geometry* metrics on both benchmarks, validating that the animated geometry using our MPM [17]-based simulation models the most accurate avatar dynamics. Ours also achieves

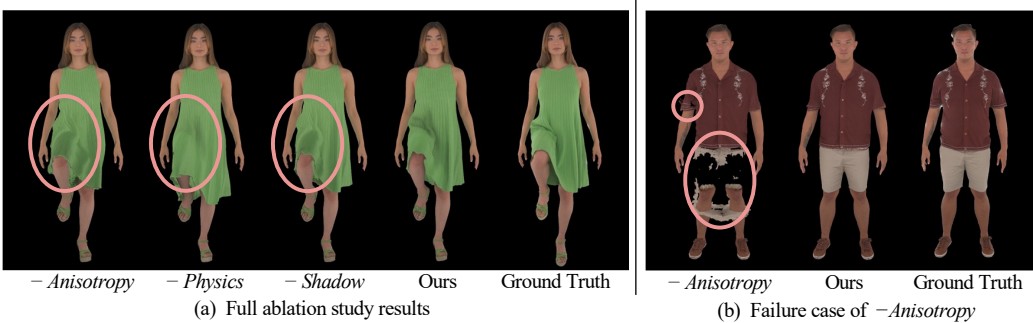

|   |   |   |   |   |   |   |   |
|---|---|---|---|---|---|---|---|
| − *Anisotropy* | − *Physics* | − *Shadow* | Ours | Ground Truth | − *Anisotropy* | Ours | Ground Truth |

(a) Full ablation study results    (b) Failure case of −*Anisotropy*

Figure 4. **Qualitative ablation study results.**

the best results across all *appearance* metrics, which shows that 3D Gaussian Splatting [22] with quasi-shadowing with the ambient occlusion prior extracted from a mesh can be an effective rendering scheme. In Fig. 3, we also show the qualitative comparisons against our most competitive baseline: PhysAvatar [78]. Our method captures complex cloth deformations (e.g., subtle wrinkles) more closely to the ground truth, and our rendering based on 3D Gaussian Splatting [22] can more effectively capture high-frequency appearance details (e.g., complex cloth patterns) than PhysAvatar based on mesh-based rendering.

**Robustness and efficiency.**    In Tab. 2, we show the simulation success rate and per-frame simulation time on the ActorsHQ [14] benchmark – compared to PhysAvatar [78]. The success rate measures the average ratio of successfully simulated frames to the total number of evaluation frames, where we mark a frame as a failure if the simulation does not terminate within 20 hours for the single frame. For this evaluation, we disabled the *manual* relaxation of the collision check for PhysAvatar, which we originally applied to prevent its simulation failures (Sec. 5.1). The success rate of PhysAvatar is 37.6% while ours is 100%, validating that ours exhibit significantly higher robustness.

For simulation time, we report the average per-frame simulation time over the test sequence. Notably, we re-applied the *manual* relaxation of the collision check in PhysAvatar, as its simulation fails to terminate without this adjustment. As shown in the table, our method achieves a simulation time of 1.1 seconds per frame, compared to 170.0 seconds for PhysAvatar, demonstrating significantly better efficiency. Note that PhysAvatar [78]'s iterative solver (C-IPC [31]) takes a long time to converge for resolving complex collisions, whereas our feed-forward MPM simulator runs much faster.

Table 2. **Simulation robustness and efficiency comparisons with PhysAvatar [78]**. Bold indicates the best scores. All scores are evaluated on the Actors-HQ dataset [14].

|   | Method | Success Rate (%) ↑ | Simulation Time (s) ↓ |
|---|---|---|---|
| A | PhysAvatar [78] | 37.6 | 170.0 |
| B | **MPMAvatar (Ours)** | **100.0** | **1.1** |

## 5.3   Ablation Study

**Anisotropic constitutive model.**   − *Anisotropy* denotes our method variant which does not use an anisotropic constitutive model [16]. As shown in Tab. 1 (Row F), this variant results in large degradation in dynamics modeling accuracy, as it does not effectively model the manifold-dependent behaviors of cloths. Few cases even have severe tearing artifacts, as shown in Fig. 4b.

**Physical parameters learning.**   − *Physics* denotes our method variant where all the physical parameters are fixed to their default values without learning. As shown in Tab. 1 (Row G), this results in suboptimal dynamics modeling accuracy, highlighting the importance of our inverse physics. In Fig. 4, we visually show that this variant leads to less accurate estimation of cloth deformations.

**Quasi-shadowing.**   − *Shadow* denotes our method variant where quasi-shadowing is not used for rendering. As shown in Tab. 1 (Row H), this results in degradation in PSNR. In Fig. 4, we qualitatively show that its rendering result exhibits significantly less realism than ours with shadowing.

## 5.4 Application: Zero-shot Scene Interaction

As an additional application, we showcase that our physics-based simulator is zero-shot generalizable to interactions with external objects unseen during training. In the right subfigures of Fig. 1, our avatar garments are naturally deformed as interacting with a chair or sand. This generalizability can be achieved as our physics-based simulator explicitly leverages the physics prior, unlike in learning-based simulators [8, 7] known to be less effective in modeling unseen dynamics. We also note that, owing to the versatility of MPM [17] in handling diverse materials, our framework supports interactions with deformable particles (e.g., sand), while simulators like C-IPC [31] are limited to mesh-based simulations. Please see Appendix A for more interaction examples of ours.

## 6 Conclusion

We presented MPMAvatar, a framework for creating 3D human avatars from multi-view videos that supports (1) physically accurate and robust animation, as well as (2) high-fidelity rendering. Our Gaussian Splat-based avatar is animated based on a carefully tailored MPM-based simulator designed for effective garment dynamics modeling, enabling physically grounded animations.

**Limitations.** Although our avatar outperformed the existing state-of-the-art physics-based avatar method [78] in both appearance and geometry, it does not support relighting as in [78]. Also, for animation, we directly followed [78] and modeled the dynamics of non-garment regions via linear blend skinning, but this can be further improved, e.g., by using strand-based simulation for hair. We refer to Appendix D for a more detailed discussion of limitations and future works.

## 7 Acknowledgement

This work was supported by NST grant (CRC21011, MSIT), IITP grant (RS-2019-II190075, RS-2023-00228996, RS-2024-00459749, RS-2025-25443318, RS-2025-25441313, MSIT) and KOCCA grant (RS-2024-00442308, MCST)

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

# A  Additional Results

In the supplementary video (available on the project page), we show additional qualitative results of our experiments.

**Qualitative Comparisons (Sec. A.1).**  We present our video results in comparison with PhysAvatar [78], a state-of-the-art physics-based avatar method. Our approach consistently achieves more accurate garment dynamics and higher rendering quality.

**Additional Qualitative Results (Sec. A.2).**  We also include additional qualitative results of our method on (1) novel pose driving and (2) zero-shot scene interactions. For novel pose driving, our method accurately models garment deformation driven by motions from the AMASS [42] dataset, which contains relatively more dynamic motions than our training motions in [14, 66]. For novel scene interactions, our method models plausible interactions between garments and a variety of materials (e.g., cushion, sand), as well as mesh-based colliders (e.g., chair, rotor) that were unseen during training. This generalization is attributed to (1) the versatility of MPM [17] and (2) our effective mesh-based collision handling method.

**Ablation Results (Sec. A.3)**  In the supplementary video, we additionally validate each of the key components of our method: (1) the constitutive model for anisotropic elastoplasticity [16], (2) physical parameter learning, (3) quasi-shadowing, and (4) rest-geometry modeling. In the fourth ablation, we directly use the canonical geometry as the rest geometry. Specifically, we optimize only Young's modulus $E$ and density $\rho$, while keeping the rest geometry parameter $\alpha$ fixed at 1. Each component is shown to be critical for accurate dynamics and appearance modeling.

**Quantitative Evaluation of Physical Plausibility (Sec. A.4).**  While we focuses on evaluating geometric and appearance fidelity which are the standard metrics in recent physics-based avatar methods [78, 69], the physical plausibility and contact accuracy are also important aspects to assess. However, quantitatively measuring these properties is challenging due to the lack of ground-truth physical annotations in real-world RGB datasets [14, 66].

Nevertheless, to provide additional insights, we report two metrics that we find to reflect physical plausibility and contact quality in practice. First, we measure the average cloth-body *Penetration Depth*, defined as the mean of $\max(0, -d)$ where $d$ is the signed distance from each cloth vertex to the SMPL-X [51] body. As shown in Tab. 3, our method significantly reduces penetration depth over $6\times$ lower than PhysAvatar [78], indicating better contact handling.

For the second metric, we adopt the *Key Physical Phenomena Detection* metric proposed in PhyGenBench [43], which computes a plausibility score using a VLM based on how well a given video aligns with physical plausibility prompts. In Tab. 3, we show that our method achieves a plausibility score closer to the ground-truth upper bound than PhysAvatar [78], indicating better alignment with the physical plausibility prompts.

Table 3. **Quantitative evaluation of physical plausibility and contact quality.** To complement standard geometry and appearance metrics, we additionally report two metrics that we find to reflect physical plausibility and contact behavior in practice. Specifically, we measure the average penetration depth (mm) and the Key Physical Phenomena Detection score [43] on the ActorsHQ [14] dataset. Our method achieves significantly lower penetration and a higher plausibility score compared to PhysAvatar [78], indicating improved contact handling and physical realism.

|   | Method | Penetration Depth (mm) ↓ | Key Physical Phenomena Detection ↑ |
|---|---|---|---|
| A | PhysAvatar [78] | 0.294 | 1.78 |
| B | **MPMAvatar (Ours)** | **0.047** | 1.83 |
| C | Ground Truth | - | **1.86** |

**Additional Ablation Results on Hyperparameters (Sec. A.5)**  To demonstrate that our pipeline remains robust across different simulation setups, we conducted an additional ablation study on the ActorsHQ [14] benchmark, where we evaluated the framework's performance while varying key hyperparameters. In Tab. 4, we observe that variations in (1) time substeps (Rows C-D), (2) physical parameter initialization (Rows E-I), and (3) mesh triangle numbers (Rows J), do not significantly affect the final performance; notably, all variants outperform PhysAvatar [78] by a clear margin across all metrics. Note that for grid resolution, the grid and particle resolutions should be roughly aligned to enable stable momentum transfer between the two representations during MPM simulation, which is also a common convention adopted in many existing MPM-based methods [4, 70]. Therefore, we omitted further ablation on this aspect.

Table 4. **Ablation study on simulation hyperparameters.** To examine the robustness of our pipeline, we ablate key simulation hyperparameters on the ActorsHQ [14] dataset. Specifically, we vary (1) the number of time substeps $N$ (Rows C–D), testing half and double our default value ($N = 400$), (2) the initialization of physical parameters $\rho$ and $E$ (Rows E–I), including $2\times$ and $0.5\times$ our defaults ($\rho = 1.0$, $E = 100$) as well as random initialization within plausible ranges, and (3) the number of mesh triangles (Row J), reducing it to one quarter of the original resolution. Across all variants, our method consistently outperforms the prior state of the art [78], showing strong robustness to hyperparameter configurations. Bold indicates the best scores, and underline indicates the second best scores.

| | Method | Geometry | | | Appearance | |
| | | CD ($\times 10^3$) ↓ | F-Score ↑ | LPIPS ↓ | PSNR ↑ | SSIM ↑ |
|---|---|---|---|---|---|---|
| A | PhysAvatar [78] | 0.55 | 92.9 | 0.035 | 30.2 | 0.957 |
| B | **MPMAvatar (Ours)** | **0.42** | **95.7** | **0.033** | 32.0 | 0.963 |
| C | Ours ($N = 800$) | **0.42** | 95.6 | **0.033** | 32.0 | 0.963 |
| D | Ours ($N = 200$) | **0.42** | 95.6 | **0.033** | 32.0 | 0.963 |
| E | Ours ($\rho = 0.5$, $E = 100$) | **0.42** | **95.7** | 0.034 | 32.0 | 0.963 |
| F | Ours ($\rho = 2.0$, $E = 100$) | **0.42** | 95.6 | 0.034 | 32.0 | 0.963 |
| G | Ours ($\rho = 1.0$, $E = 50$) | **0.42** | 95.6 | 0.034 | 32.0 | 0.963 |
| H | Ours ($\rho = 1.0$, $E = 200$) | **0.42** | **95.7** | 0.034 | 32.0 | 0.963 |
| I | Ours (randomly initialized) | 0.43 | 95.5 | **0.033** | 32.0 | 0.963 |
| J | Ours ($0.25\times$ triangles) | 0.44 | 95.4 | **0.033** | **32.1** | **0.964** |

Table 5. **Quantitative comparison against concurrent baselines on the ActorsHQ [14] dataset.** Bold and underlined values indicate the best and second-best scores, respectively. Our method consistently outperforms recent baselines across all geometry and appearance metrics, highlighting the advantage of physics-based simulation.

| | Method | Geometry | | | Appearance | |
| | | CD ($\times 10^3$) ↓ | F-Score ↑ | LPIPS ↓ | PSNR ↑ | SSIM ↑ |
|---|---|---|---|---|---|---|
| A | Gaussian Garments [55] | 2.39 | 86.5 | 0.042 | 29.5 | 0.959 |
| B | MMLPHuman [74] | 0.47 | 94.9 | 0.039 | 29.3 | 0.954 |
| C | **MPMAvatar (Ours)** | **0.42** | **95.7** | **0.033** | **32.0** | **0.963** |

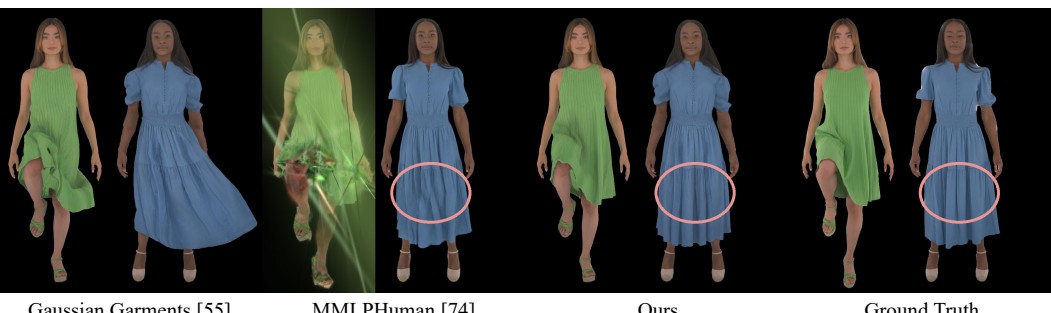

| Gaussian Garments [55] | MMLPHuman [74] | Ours | Ground Truth |

Figure 5. **Qualitative comparison against concurrent baselines on the ActorsHQ [14] dataset.** We compare our method with two recent concurrent methods: Gaussian Garments [55] and MMLPHuman [74]. Gaussian Garments [55] struggles to produce physically accurate deformations, while MMLPHuman [74] exhibits unnatural surface artifacts or discontinuities under challenging poses. In contrast, our method yields more realistic and plausible garment dynamics and geometry.

**Comparison with Additional Concurrent SOTA Baselines (Sec. A.6)** To further validate our approach, we additionally compare our method against two recent concurrent state-of-the-art avatar reconstruction methods: **Gaussian Garments** [55] and **MMLPHuman** [74]. Gaussian Garments replaces explicit simulation with a learned dynamics module based on graph neural networks, while MMLPHuman models geometry and appearance solely as a function of pose using multiple MLPs.

For both Gaussian Garments [55] and MMLPHuman [74] experiments, we ensure a fair comparison by aligning key experimental components with our method. For Gaussian Garments [55], we replace our MPM-based simulation with the learning-based garment simulator from ContourCraft [7], which serves as the simulation backend of Gaussian Garments [55], while keeping the tracked meshes and rendering pipeline identical to ours. Since Gaussian Garments [55] models the human body using only SMPL-X [51] without explicit body appearance modeling, we also use our body modeling setup to ensure a consistent evaluation environment. For MMLPHuman [74], we follow the official

implementation and evaluation protocol, and additionally use the same template mesh as in our method.

As shown in Fig. 5 and Tab. 5, our method outperforms both baselines across all geometry and appearance metrics. The learned simulator in Gaussian Garments [55] struggles to capture physical laws under our setting, where physical parameters must be estimated from only one second of motion, leading to high geometric error. Meanwhile, MMLPHuman [74] lacks explicit surface modeling and physical understanding, producing unrealistic surface artifacts or broken geometry when encountering unseen poses. These results demonstrate the advantage of our physics-based modeling pipeline for reconstructing accurate and plausible dynamic avatars.

# B  Implementation Details

## B.1  Physics-Based Dynamics Modeling

**Collision handling.** In Alg. 1, we outline our collision handling algorithm. After parameter initialization (lines 1-3), we iterate over every face $f$ in the collider mesh and transfer its velocity $\mathbf{v}_f$ and normal $\mathbf{n}_f$ to nearby grid nodes based on B-Spline weights (lines 5-11). This yields the extended velocity $\mathbf{v}_i^c$ and normal $\mathbf{n}_i^c$ of the collider at grid node $i$. Then, given the velocity of the simulating object $\mathbf{v}_i$ at grid node $i$, if the vector $\mathbf{v}_i - \mathbf{v}_i^c$ points inward, we project out the normal component — keeping only the tangential part — to model collision (lines 13–24). Note that this entire procedure runs in $O(N_f)$ time, as collision checks become simple B-Spline weight lookups rather than costly level-set queries at all grid nodes.

---

**Algorithm 1** Collision Handling

---

1: **for** each $i$ in a set of grid node indices **do**
2:     $\mathbf{v}_i^c, \mathbf{n}_i^c \leftarrow \mathbf{0}, \mathbf{0}$;                                     $\triangleright$ Initialize the zero-value grids for collider
3: **end for**
4:
5: **for** each $f$ in a set of collider mesh faces **do**
6:     **for** $i$ in a set of neighboring grid nodes of $\mathbf{x}_f$ **do**
7:         $w_{if}^c \leftarrow Bspline(\mathbf{x}_f, \mathbf{x}_i)$;               $\triangleright$ Compute interpolation weights using B-spline kernel
8:         $\mathbf{v}_i^c \leftarrow \mathbf{v}_i^c + w_{if}^c \mathbf{v}_f$;
9:         $\mathbf{n}_i^c \leftarrow \mathbf{n}_i^c + w_{if}^c \mathbf{n}_f$;
10:     **end for**
11: **end for**
12:
13: **for** each $i$ in a set of grid node indices **do**
14:     $w_i^c \leftarrow \sum_f w_{if}^c$
15:     **if** $w_i^c > 0$ **then**                           $\triangleright$ Detect whether a collision has occurred
16:         $\mathbf{v}_i^c \leftarrow \mathbf{v}_i^c / w_i^c$;
17:         $\mathbf{n}_i^c \leftarrow \mathbf{n}_i^c / \|\mathbf{n}_i^c\|$;
18:         $\mathbf{v}_i^{rel} \leftarrow \mathbf{v}_i - \mathbf{v}_i^c$;                $\triangleright$ Transform velocities into the collider's reference frame
19:         **if** $\mathbf{v}_i^{rel} \cdot \mathbf{n}_i^c < 0$ **then**       $\triangleright$ Check if the relative velocity points inward toward the collider
20:             $\mathbf{v}_i^{rel} \leftarrow \mathbf{v}_i^{rel} - (\mathbf{v}_i^{rel} \cdot \mathbf{n}_i^c)\mathbf{n}_i^c$;     $\triangleright$ Project relative velocity onto the collider's tangent space
21:         **end if**
22:         $\mathbf{v}_i \leftarrow \mathbf{v}_i^{rel} + \mathbf{v}_i^c$                    $\triangleright$ Transform velocities back into world frame
23:     **end if**
24: **end for**

---

**Physical parameters learning.** As discussed in Sec. 4.3.1 in the paper, we optimize Young's modulus $E$, density $\rho$, and rest geometry parameter $\alpha$ by simulating the first-frame canonical mesh $\mathcal{M}_1 = (\mathbf{V}_1, \mathbf{F})$ and minimizing the vertex-wise $L_2$ error with respect to the tracked meshes $(\mathcal{M}_i)_{i=2,\dots,T}$, where $\mathcal{M}_i = (\mathbf{V}_i, \mathbf{F})$.

In particular, let the initial mesh vertices be $\hat{\mathbf{V}}_1 = \mathbf{V}_1$. Then, for each subsequent frame, we obtain the simulated mesh vertices via

$$\hat{\mathbf{V}}_{i+1} = \text{MPM}(\hat{\mathbf{V}}_i, \mathbf{V}_1, \mathbf{F}, \mathcal{P}), \tag{2}$$

where $\mathcal{P} = (E, \nu, \gamma, \kappa, \rho, \alpha)$ are our physical parameters.

Here, the canonical vertices $\mathbf{V}_1$ and faces $\mathbf{F}$ provide the fixed mesh topology used to compute edge vectors and material directions, which in turn define the deformation gradients for the anisotropic constitutive model inside the MPM simulation.

We then perform gradient-based optimization for $\rho$, $E$ and $\alpha$, such that they minimize the loss

$$\mathcal{L}_{\text{phys}}(\mathcal{P}) = \sum_{i=2}^{T} \|\hat{\mathbf{V}}_i - \mathbf{V}_i\|^2. \tag{3}$$

Here, each parameter's gradient is approximated using finite differences – following PhysAvatar [78]. For example, the gradient with respect to density $\rho$ is computed as

$$\frac{\partial \mathcal{L}_{\text{phys}}}{\partial \rho} \approx (\mathcal{L}_{\text{phys}}(E, \nu, \gamma, \kappa, \rho + \Delta\rho, \alpha) - \mathcal{L}_{\text{phys}}(E, \nu, \gamma, \kappa, \rho, \alpha))/\Delta\rho, \tag{4}$$

where $\Delta\rho$ is the perturbation size.

**Training details.**

Our MPM simulation uses a time step of $\Delta t = 0.04$ with $N = 400$ substeps and a grid resolution of 200. We optimize the physical parameters over 200 iterations using the Adam optimizer. For finite-difference gradient estimation, the perturbation sizes are set to $\Delta\rho = 0.05$, $\Delta E = 5$, and $\Delta\alpha = 0.005$. The corresponding learning rates are 0.01 for $\rho$, 0.3 for $E$, and 0.01 for $\alpha$. All parameters are initialized as $\rho = 1.0$, $E = 100$, and $\alpha = 1.0$ for physical parameter learning, while $\nu$, $\gamma$, and $\kappa$ are fixed at their default values of 0.3, 500, and 500, respectively.

Following PhysAvatar [78], we adopt a stage-wise training scheme: the physical parameters are optimized first and the tracked meshes remain fixed throughout. Appearance learning is then performed independently based on the same tracked meshes (see Sec. 4.3).

**Simulation time and computing resource.** As noted in the main paper (Sec. 5.2), our simulation runs at approximately 1.1 seconds per frame on a single NVIDIA GeForce RTX 4090.

## B.2 Appearance Learning

For the appearance learning (Sec. 4.3.2 in the paper), we leverage the dense temporal correspondences from the tracked meshes $(\mathcal{M}_i)_{i=1\ldots T}$ to transform the canonical Gaussians $\mathcal{G}$ (defined in the first frame) into each subsequent frame. Following [54], we compute the transformations that carries every Gaussian from its parent triangle in the canonical mesh to the corresponding triangle in the target frame. This allows us to render all training frames $[1, \ldots, T]$ using a single shared appearance model $\mathcal{G}$, such that it can be learned jointly from all input views and frames by minimizing:

$$\mathcal{L}_{\text{app}} = \mathcal{L}_{\text{rgb}} + \lambda_{\text{p}}\mathcal{L}_{\text{position}} + \lambda_{\text{s}}\mathcal{L}_{\text{scaling}}. \tag{5}$$

Here, $\mathcal{L}_{\text{rgb}}$ measures the photometric discrepancy between the rendered and the ground truth images over all frames and views, and is defined as a weighted sum of L1 loss $\mathcal{L}_1$, SSIM [67] loss $\mathcal{L}_{\text{SSIM}}$, and LPIPS [75] loss $\mathcal{L}_{\text{LPIPS}}$:

$$\mathcal{L}_{\text{rgb}} = \lambda_1\mathcal{L}_1 + \lambda_{\text{SSIM}}\mathcal{L}_{\text{SSIM}} + \lambda_{\text{LPIPS}}\mathcal{L}_{\text{LPIPS}}. \tag{6}$$

In Eq. 5, $\mathcal{L}_{\text{position}} = \|\max(\mu, \epsilon_{\text{p}})\|_2$ and $\mathcal{L}_{\text{scaling}} = \|\max(s, \epsilon_{\text{s}})\|_2$ regularize the location offset $\mu$ and the scale $s$ of each Gaussian not to exceed the thresholds $\epsilon_{\text{p}} = 1.0$ and $\epsilon_{\text{s}} = 0.6$. This is to encourage the Gaussians to remain closely aligned with their parent triangle structures.

When optimizing the parameters of $\mathcal{G}$, we follow the standard 3DGS optimization procedure [22] and employ adaptive density control to increase the number of Gaussians in regions with high reconstruction error. For the loss weighting hyperparameters, we set $\lambda_1 = 0.8$, $\lambda_{\text{SSIM}} = 0.2$, $\lambda_{\text{LPIPS}} = 0.2$, $\lambda_{\text{p}} = 1.0$, and $\lambda_{\text{s}} = 1.0$.

# C  Anisotropic Constitutive Model

In this section, we provide additional details on the anisotropic constitutive model used in our MPM simulation framework, to complement the explanation provided in Sec. 3 and Sec. 4.2.1 of the main paper. This elaboration aims to help readers better understand how our method captures direction-dependent garment dynamics.

As explained in Sec. 3, we employ the Material Point Method (MPM) [17] to simulate the evolution of deformable objects by solving two governing equations: conservation of mass and conservation of momentum (Eq. 1). Among these, conservation of momentum governs the time evolution of velocity, and its simulation hinges on the computation of the Cauchy stress tensor $\boldsymbol{\sigma}$. This stress tensor depends on the deformation gradient $\mathbf{F}$ and the strain-energy density function $\psi$ via

$$\boldsymbol{\sigma} = \frac{1}{\det(\mathbf{F})} \frac{\partial \psi}{\partial \mathbf{F}} \mathbf{F}^\top,$$

where $\psi$ is defined by a material-specific constitutive model.

As introduced in Sec. 4.2.1, we adopt the anisotropic constitutive model proposed by Jiang et al. [16], which is particularly well-suited for modeling thin, codimensional structures like garments. This model captures how cloth exhibits strong resistance to compression and shearing along the surface normal while remaining flexible along in-plane directions.

To compute the deformation gradient $\mathbf{F}$ at each particle, the model uses local material directions derived from a Lagrangian mesh. Specifically,

$$\mathbf{F} = \mathbf{d}\mathbf{D}^{-1},$$

where $\mathbf{D} = [\mathbf{D}_1, \mathbf{D}_2, \mathbf{D}_3]$ denotes the canonical (undeformed) material directions and $\mathbf{d} = [\mathbf{d}_1, \mathbf{d}_2, \mathbf{d}_3]$ denotes the corresponding deformed directions. Since the strain energy function $\psi$ must be invariant under rotations, the model applies QR decomposition $\mathbf{F} = \mathbf{QR}$ and reparameterizes the energy as a function of the upper-triangular matrix $\mathbf{R}$:

$$\psi(\mathbf{F}) = \hat{\psi}(\mathbf{R}) = \hat{\psi}_{\text{normal}} + \hat{\psi}_{\text{shear}} + \hat{\psi}_{\text{in-plane}},$$

where each term independently penalizes a specific type of deformation.

The normal component penalizes compression along the surface normal:

$$\hat{\psi}_{\text{normal}}(\mathbf{R}_{33}|\kappa) = \begin{cases} \frac{\kappa}{3}(1 - \mathbf{R}_{33})^3 & \text{if } \mathbf{R}_{33} \leq 1, \\ 0 & \text{otherwise,} \end{cases}$$

reflecting the assumption that cloth is typically surrounded by air and thus can freely expand but should resist compression.

The shear component penalizes off-diagonal shear deformation between in-plane and normal directions:

$$\hat{\psi}_{\text{shear}}(\mathbf{R}_{13}, \mathbf{R}_{23}|\gamma) = \frac{\gamma}{2}(\mathbf{R}_{13}^2 + \mathbf{R}_{23}^2),$$

which stabilizes the material frame by discouraging bending or tilting out of plane.

The in-plane component models isotropic stretching within the tangent plane using a fixed-corotated formulation:

$$\hat{\psi}_{\text{in-plane}}(\mathbf{R}_{11}, \mathbf{R}_{12}, \mathbf{R}_{22}|E, \nu) = \frac{E}{2(1+\nu)}((\sigma_1 - 1)^2 + (\sigma_2 - 1)^2) + \frac{E\nu}{2(1+\nu)(1-2\nu)}(\sigma_1\sigma_2 - 1)^2,$$

where $\sigma_1, \sigma_2$ are the singular values of the in-plane matrix

$$\mathbf{R}^{2\times 2} = \begin{bmatrix} \mathbf{R}_{11} & \mathbf{R}_{12} \\ 0 & \mathbf{R}_{22} \end{bmatrix}.$$

This overall anisotropic formulation provides the strain-energy density $\psi$ needed to compute the stress tensor $\boldsymbol{\sigma}$ in the momentum equation (Eq. 1b), thereby enabling our simulator to capture realistic garment behavior with directionally varying stiffness. This detailed model plays a central role in achieving the accurate and physically plausible dynamics demonstrated in our results.

# D   Limitations and Future Work

While our method achieves state-of-the-art performance in both appearance and physical dynamics modeling, we acknowledge several limitations and outline potential directions for future work.

**Scalability of Finite-Difference Optimization.**   Our physical parameter optimization adopts a finite-difference scheme, which scales linearly with the number of parameters. While this remains practical for our current setting, where per-garment material parameters suffice due to limited intra-garment heterogeneity, extending to fine-grained parameterizations (e.g., per-vertex) would increase computational cost. As a mitigation strategy, incorporating differentiable simulators [37, 33] may improve scalability in future applications.

**Relighting.**   Our current framework does not support relightable rendering. However, recent methods [19, 57] have proposed relighting-aware extensions for Gaussian avatars, and our hybrid representation is compatible with such techniques. We consider this a promising direction to further enhance rendering realism.

**Occlusion-Aware Generalization.**   Our pipeline directly optimizes appearance only in regions visible in the multi-view training frames. Consequently, when previously occluded or unseen parts (e.g., the back side of the avatar) become visible under novel poses or viewpoints, rendering quality may degrade. Recent works have explored generative priors to inpaint unobserved regions [26], or diffusion-based view synthesis to generate pseudo multi-view supervision from monocular videos [63]. Incorporating such approaches into our pipeline could improve generalization to occluded or unseen regions.

# E   Societal Impact

Our method enables physically accurate dynamic human avatar reconstruction from multi-view videos, supporting a wide range of applications in virtual reality, digital fashion, and entertainment. However, the capability to generate lifelike avatars also introduces potential risks, such as the misuse of the technology for creating deepfakes or other forms of deceptive content. When publishing our code, we will consider embedding traceable digital watermarks or developing authentication mechanisms to ensure the responsible use of generated avatars.

