# OpenReview forum: "MPMAvatar: Learning 3D Gaussian Avatars with Accurate and Robust Physics-Based Dynamics"
_NeurIPS.cc/2025/Conference — NeurIPS 2025 poster_

### Official Review · Reviewer_xXua · 2025-06-23

**Clarity:** 2
**Significance:** 3
**Originality:** 3
**Rating:** 4
**Confidence:** 4

**Summary:**

This paper introduces MPMAvatar, a method for learning physically accurate and robust 3D garment dynamics from multi-view video using the Material Point Method (MPM). The approach simulates only the garments using an anisotropic elastoplastic constitutive model, while treating the human body and external scene objects as colliders. A small set of physical parameters—including Young’s modulus, density, and rest shape scaling—are optimized via finite-difference methods to match tracked garment meshes. For appearance, the method leverages 3D Gaussian Splatting (3DGS) with a shared canonical representation and quasi-shadowing to produce photorealistic rendering across views. Compared to PhysAvatar, MPMAvatar demonstrates superior accuracy, simulation robustness, generalization to novel interactions, and rendering quality. It also runs orders of magnitude faster than an IPC-based baseline.

**Questions:**

1. The method first constructs a tracked mesh from multi-view videos. Then, it performs MPM material optimization, followed by appearance optimization. Is this a stage-wise optimization, i.e., MPM and appearance are optimized separately? Also, will either of these 2 optimizations modify the underlying tracked-mesh?

2. Regarding the Geometry metric reported in Table 1, are the inputs to PhysAvatar and MPMAvatar exactly the same? Does rendering have any effect on Geometry metrics in this experiment? I ask this because to confirm the effectiveness of MPM, we should rule out any other factors (e.g., 3DGS rendering) that could benefit Geometry metrics.

3. In the bottom row of the right subfigure of Fig. 2, it is unclear what "Grid Velocity", "Projected Velocity", and "Final Velocity" are.

5. The term "edge vector" (line 264) is introduced abruptly without a formal definition, which can hinder clarity. Further, it is unclear why this edge vector decomposition is crucial for learning the scale parameter, as the paper does not introduce anywhere how edge vectors are involved in an anisotropic constitutive model. Simply referring to Jiang et al. is not enough; there should at least be a section that builds connections between garment mesh topologies and the actual MPM function. Currently, only Eq. (1) in the Supp. Mat. treats MPM as a function of current vertices and physical parameters, but provides no more details. In this context, it is impossible to judge why the "edge vector decomposition" technique is helpful. Also, if MPM takes only the vertices of the garment mesh as input, how do "edge vectors" affect MPM at all?

**Ethical Concerns:**

["NO or VERY MINOR ethics concerns only"]

**Final Justification:**

The authors addressed my concerns, and I would like to keep my positive scores.

**Limitations:**

Yes

**Quality:**

3

**Strengths And Weaknesses:**

Strengths:
1. By using MPM with a novel mesh-based collision handler, the method avoids simulation crashes common in PhysAvatar and achieves a 100% success rate across sequences.

2. The method supports physically plausible cloth behavior (e.g., folds, collisions) with \~1.1s/frame simulation time—substantially faster than PhysAvatar (\~170s/frame).

3. MPMAvatar handles unseen pose sequences and novel scene-object interactions (e.g., garments contacting cushions or sand) without retraining, attributed to the physically grounded formulation.

4. The use of 3D Gaussian Splatting, combined with quasi-shadowing, results in visually superior renderings compared to mesh-based avatars.

5.  The supplementary video and ablation studies isolate the contributions of anisotropic modeling, rest geometry correction, and quasi-shadowing.

Weaknesses (please see the Questions section for details):
1. Although there are some high-level descriptions in the Supp. Mat., the learning process is unclear,

2. The experimental setup also has some clarity issue, which makes the effectiveness of using MPM questionable.

3. Fig.2, especially the collision handling part, is confusing. While the B-spline part is explained sufficiently in Sec. 4.2.2 (main paper) and Alg. 1 (Supp. Mat.), the bottom row needs more clarification.

4. Some quantities, such as edge vectors, are brought up without context and need more clarification.

5. The Quasi-shadow modeling gives only marginal improvement.

---

> ### Author Rebuttal · Authors · 2025-07-30
>
> # General Response
>
> We thank the reviewers for their constructive feedback and recognizing the **novelty of our collision handling method** (gjLz, 1QGS), **strong quantitative and qualitative results in dynamics, rendering quality, and efficiency** (xXua, gjLz, 1QGS, 1m12), as well as our method’s **zero-shot generalizability to unseen physical interactions** (xXua, 1QGS).
>
> In response to the reviewers’ feedback, we have made several additions to strengthen the paper. Specifically, we conducted three new experiments:
> * **penetration depth analysis** for evaluating contact handling,
> * a **VLM-based assessment** of physical plausibility using PhyGenBench [R1], and
> * **additional ablations** on key simulation hyperparameters.
>
> We also updated Tab. 1 to include comparisons with two recent state-of-the-art baselines [43, R2]. We hope these revisions address the reviewers’ concerns and welcome any further suggestions.
>
> **Table R1:** Quantitative comparison results including new baselines
> | Method | CD ($\times 10^{3}$) (↓) | F-Score (↑) | LPIPS (↓) | PSNR (↑) | SSIM (↑) |
> |----------|----------|----------|----------|----------|----------|
> | MMLPHuman [R2] | 0.47 | 94.9 | 0.039 | 29.3 | 0.954 |
> | Gaussian Garments [43] | 2.39 | 86.5 | 0.042 | 29.5 | 0.959 |
> | PhysAvatar [61] | 0.55 | 92.9 | 0.035 | 30.2 | 0.957 |
> | Ours | **0.42** | **95.7** | **0.033** | **32.0** | **0.963** |
>
> [R1] Meng *et al.* Towards World Simulator: Crafting Physical Commonsense-Based Benchmark for Video Generation, In ICML, 2025.
>
> [R2] Zhan *et al.* Real-time High-fidelity Gaussian Human Avatars with Position-based Interpolation of Spatially Distributed MLPs. In CVPR, 2025.
>
> ---
>
> # Response to Reviewer xXua
>
> > **Weakness 1.** "Although there are some high-level descriptions in the Supp. Mat., the learning process is unclear"
>
> > **Questions 1.** "The method first constructs a tracked mesh from multi-view videos. Then, it performs MPM material optimization, followed by appearance optimization. Is this a stage-wise optimization, i.e., MPM and appearance are optimized separately? Also, will either of these 2 optimizations modify the underlying tracked-mesh?"
>
> **Reply:** We clarify that we perform stage-wise optimization, where the physical parameters and appearance are optimized separately. Throughout these stages, we do not modify the tracked meshes obtained during data pre-processing, as they serve as pseudo ground truth to “supervise the subsequent learning stages” (L239–240 in the paper). Specifically, the tracked meshes are used to compute (1) vertex-level L2 losses during physical parameter optimization (L33–36 in the supplementary) and (2) photometric losses during appearance optimization (L53–55 in the supplementary). We also note that our stage-wise optimization and use of tracked meshes as pseudo ground truths were primarily motivated by the common practice in prior physics-based avatar work [61]. We thank you for the suggestion and will further clarify these points in the revision.
>
> > **Weakness 2.** "The experimental setup also has some clarity issue, which makes the effectiveness of using MPM questionable."
>
> > **Questions 2.** "Regarding the Geometry metric reported in Table 1, are the inputs to PhysAvatar and MPMAvatar exactly the same? Does rendering have any effect on Geometry metrics in this experiment? I ask this because to confirm the effectiveness of MPM, we should rule out any other factors (e.g., 3DGS rendering) that could benefit Geometry metrics."
>
> **Reply:** We clarify that, when measuring the Geometry metrics (Tab. 1 in the paper), (1) the inputs to PhysAvatar [61] and our method are the same, and (2) rendering quality does not affect these metrics. As described in L316–317 in the main paper, these Geometry metrics (Chamfer Distance and F-Score) are computed between the simulated and ground-truth meshes, independent of the rendering strategy. While our avatar uses a hybrid representation (mesh + 3D Gaussians), only the mesh component is used for geometry evaluation, ensuring a fair comparison.
>
> > **Weakness 3.** "Fig.2, especially the collision handling part, is confusing. While the B-spline part is explained sufficiently in Sec. 4.2.2 (main paper) and Alg. 1 (Supp. Mat.), the bottom row needs more clarification."
>
> > **Questions 3.** "In the bottom row of the right subfigure of Fig. 2, it is unclear what "Grid Velocity", "Projected Velocity", and "Final Velocity" are."
>
> **Reply:** The bottom row of Fig. 2 follows the standard collision handling procedure in MPM [17], where:
>
> * **Grid Velocity** denotes the original velocity of the garment at each colliding grid node (L224-226);
>
> * **Relative Velocity** denotes the garment’s velocity relative to the corresponding collider grid (L224-226);
>
> * **Projected Velocity** denotes the relative velocity after removing the inward normal component based on the body’s surface normal, to prevent penetration (L226); and
>
> * **Final Velocity** is the projected relative velocity converted back into the world-frame garment velocity after projection (L226-227).
>
> While we provided a high-level description of this process in Sec. 4.2.2 of the main paper, as well as in Alg. 1 and L23–30 of the supplementary material, we will make sure to explicitly clarify the figure notations and strengthen the connection between the figure and the text in the revision.
>
> > **Weakness 4.** "Some quantities, such as edge vectors, are brought up without context and need more clarification."
>
> > **Questions 4.** "The term "edge vector" (line 264) is introduced abruptly without a formal definition, which can hinder clarity. Further, it is unclear why this edge vector decomposition is crucial for learning the scale parameter, as the paper does not introduce anywhere how edge vectors are involved in an anisotropic constitutive model. Simply referring to Jiang et al. is not enough; there should at least be a section that builds connections between garment mesh topologies and the actual MPM function. Currently, only Eq. (1) in the Supp. Mat. treats MPM as a function of current vertices and physical parameters, but provides no more details. In this context, it is impossible to judge why the "edge vector decomposition" technique is helpful. Also, if MPM takes only the vertices of the garment mesh as input, how do "edge vectors" affect MPM at all?"
>
> **Reply:** Thank you for your comment — we will make sure to include the formal definition of the edge vectors in the revision. We clarify that the edge vectors (L264) refer to the side vectors of each triangle in the canonical mesh. The orthogonal decomposition of these edge vectors is necessary to apply the scale parameters $\alpha$ along the axis aligned with the gravity direction (and not along other directions; L267-268), as our goal was “to compensate for gravity-induced deformation” (L263–264) when approximating the rest geometry.
>
> Regarding the connections between the mesh topology and the MPM simulation, we clarify that the mesh topology is used to compute the material directions $\mathbf{D}$ (L191), which are then used to compute the deformation gradients $\mathbf{F}$ in the anisotropic constitutive model (L190–197). Based on these deformation gradients, MPM [17] “internally computes forces relative to the object’s rest geometry” (L257–258). Initially, we only provided high-level descriptions, as these connections are mostly established in the existing MPM [17] and the anisotropic constitutive model [16] that we adopted. However, we agree that this hinders clarity for readers unfamiliar with these works. Following your comment, we will add a separate section in the supplementary to explain in detail how the edge vectors are used to compute deformations in these existing methods, and we will also modify the simplified Eq. (1) in the supplementary to explicitly take the mesh topology.
>
> > **Weakness 5.** "The Quasi-shadow modeling gives only marginal improvement."
>
> **Reply:** While we agree that quasi-shadow modeling yields only slight improvements in quantitative metrics, we believe its qualitative impact is non-negligible, particularly in garment regions with large deformations or self-occlusions. As shown in our ablation study results (Fig. 4(a) in the paper and the supplementary video), removing quasi-shadowing leads to noticeably less realistic visual results. In photometric avatar rendering applications, where perceptual plausibility plays a central role, we believe such qualitative improvements remain meaningful, even when commonly used pixel- or feature-level metrics show modest gains.

---

> > ### Comment · Reviewer_xXua · 2025-08-04
> >
> > Thanks for the detailed response! My questions are well addressed and I have no problem for accepting the paper now.

---

> > > ### Author Response · Authors · 2025-08-04
> > >
> > > Thank you for your response, and once again, for your constructive reviews, which have helped improve our manuscript.
> > >
> > > We’re glad to hear that our rebuttal has addressed your questions. If your concerns have been sufficiently resolved, we would greatly appreciate it if you could kindly consider updating your rating.
> > >
> > > Thank you again for engaging in the discussion with us.

---

### Official Review · Reviewer_gjLz · 2025-06-23

**Clarity:** 2
**Significance:** 3
**Originality:** 3
**Rating:** 4
**Confidence:** 4

**Summary:**

This paper presents MPMAvatar, a hybrid framework for learning physically plausible 3D human avatars from multi-view videos. It integrates a Material Point Method (MPM)-based cloth simulator with 3D Gaussian Splatting for high-fidelity rendering. Compared to prior works (e.g., PhysAvatar), the proposed system achieves more accurate garment dynamics, improved visual quality, and greater robustness under challenging animation inputs, including zero-shot scene interactions.

Key contributions include:

Introducing a tailored MPM pipeline with an anisotropic constitutive model to model cloth dynamics.

Proposing a mesh-based collision handling algorithm for robust simulation against SMPL-X driven body inputs.

Overall, I like the paper even I still have several concerns and the score will change according to the discussion and other reviews.

**Questions:**

Please refer to the weakness part.

**Ethical Concerns:**

["NO or VERY MINOR ethics concerns only"]

**Final Justification:**

After the rebuttal, most of my concerns are resolved, this I will keep my positive score.

**Limitations:**

Limited discussion of limitations: While the authors briefly mention in the conclusion that they do not support relighting or strand-based simulation, a more thorough analysis of the method's limitations is missing.

**Paper Formatting Concerns:**

No formatting concerns.

**Quality:**

3

**Strengths And Weaknesses:**

**Strengths:**

Technically sound and novel: Tailoring MPM with anisotropic elasticity and mesh-aware collisions enables robust garment simulation that surpasses C-IPC-based avatars in both quality and efficiency.

Efficient simulation: The proposed MPM implementation reduces simulation time alot compared to PhysAvatar due to the use of MPM.

**Weaknesses:**

Single-material assumption: The method assumes the entire garment shares a single set of material parameters (e.g., a global Young’s modulus E). This limits its applicability to heterogeneous clothing composed of materials with different physical properties (e.g., jeans + silk top). No support is provided for segment-wise or per-region physical parameter estimation.

Discussion of simulation parameters: MPM’s accuracy and stability are highly sensitive to parameter initializaton, particle numbers, grid resolution and time step, yet the paper provides few discussion or empirical justification for their selection. These are critical hyperparameters in practical deployment.

Lack of failure case analysis: The paper primarily focuses on showcasing positive results but does not include any failure cases or challenging scenarios where the method might break down. For instance: (1) Are there cases where the simulation fails due to numerical instabilities or incorrect body-garment collisions? (2) Are certain garment types (e.g., multilayered skirts, long coats) poorly handled?

---

> ### Author Rebuttal · Authors · 2025-07-30
>
> # General Response
>
> We thank the reviewers for their constructive feedback and recognizing the **novelty of our collision handling method** (gjLz, 1QGS), **strong quantitative and qualitative results in dynamics, rendering quality, and efficiency** (xXua, gjLz, 1QGS, 1m12), as well as our method’s **zero-shot generalizability to unseen physical interactions** (xXua, 1QGS).
>
> In response to the reviewers’ feedback, we have made several additions to strengthen the paper. Specifically, we conducted three new experiments:
> * **penetration depth analysis** for evaluating contact handling,
> * a **VLM-based assessment** of physical plausibility using PhyGenBench [R1], and
> * **additional ablations** on key simulation hyperparameters.
>
> We also updated Tab. 1 to include comparisons with two recent state-of-the-art baselines [43, R2]. We hope these revisions address the reviewers’ concerns and welcome any further suggestions.
>
> **Table R1:** Quantitative comparison results including new baselines
> | Method | CD ($\times 10^{3}$) (↓) | F-Score (↑) | LPIPS (↓) | PSNR (↑) | SSIM (↑) |
> |----------|----------|----------|----------|----------|----------|
> | MMLPHuman [R2] | 0.47 | 94.9 | 0.039 | 29.3 | 0.954 |
> | Gaussian Garments [43] | 2.39 | 86.5 | 0.042 | 29.5 | 0.959 |
> | PhysAvatar [61] | 0.55 | 92.9 | 0.035 | 30.2 | 0.957 |
> | Ours | **0.42** | **95.7** | **0.033** | **32.0** | **0.963** |
>
> [R1] Meng *et al.* Towards World Simulator: Crafting Physical Commonsense-Based Benchmark for Video Generation, In ICML, 2025.
>
> [R2] Zhan *et al.* Real-time High-fidelity Gaussian Human Avatars with Position-based Interpolation of Spatially Distributed MLPs. In CVPR, 2025.
>
> ---
>
> # Response to Reviewer gjLz
>
> > **Weakness 1.** "Single-material assumption: The method assumes the entire garment shares a single set of material parameters (e.g., a global Young’s modulus E). This limits its applicability to heterogeneous clothing composed of materials with different physical properties (e.g., jeans + silk top). No support is provided for segment-wise or per-region physical parameter estimation."
>
> **Reply:** We clarify that we use a single-material assumption **per garment**; thus, top and bottom garments (e.g., jeans and a silk top) have different material parameters. Indeed, the datasets [14, 51] used in our experiments contain several subjects wearing two types of garments (e.g., Fig. 3(b)), and we performed per-garment optimization in those cases—directly following the practice used in the current state-of-the-art method (PhysAvatar [61]). We will further clarify these points in the revision.
>
> Although we did not initially consider more fine-grained (e.g., per-vertex) material parameterization—as the datasets used in our experiments do not exhibit significant material heterogeneity within the same garment—we agree that this is an important direction for future work to enable more expressive garment modeling. We will include this discussion in the revision as well.
>
> > **Weakness 2.** "Discussion of simulation parameters: MPM’s accuracy and stability are highly sensitive to parameter initializaton, particle numbers, grid resolution and time step, yet the paper provides few discussion or empirical justification for their selection. These are critical hyperparameters in practical deployment."
>
> **Reply:** We thank the reviewer for the insightful comment. To assess the sensitivity of our framework to these hyperparameters, we conducted an additional ablation study on the ActorsHQ [14] benchmark (Actor 01), where we evaluated the framework’s performance while varying key hyperparameters. In Tab. R2, we observe that variations in (1) physical parameter initialization, (2) particle numbers, and (3) time substeps do not significantly affect the final performance; notably, all variants outperform PhysAvatar [61] by a clear margin across all metrics. Note that for grid resolution, the grid and particle resolutions should be roughly aligned to enable stable momentum transfer between the two representations during MPM simulation, which is also a common convention adopted in many existing MPM-based methods [4, 54]. Therefore, we omitted further ablation on this aspect.
>
> While we primarily presented our ablation results on Actor 01 (as mentioned above) in the rebuttal due to time constraints, we will also run experiments on all sequences and include the complete results in the revision. We again thank the reviewer for the constructive feedback.
>
> **Table R2:** $N$ denotes the number of time substeps between two consecutive frames. $\rho$ and $E$ represent the initial values of density and Young’s modulus used for physical parameter learning, respectively.
> | Method | CD ($\times 10^{3}$) (↓) | F-Score (↑) | LPIPS (↓) | PSNR (↑) | SSIM (↑) |
> |----------|----------|----------|----------|----------|----------|
> | Ours (Full) | **0.38** | 95.7 | **0.025** | **34.1** | 0.964 |
> | Ours ($N=800$) | 0.39 | 95.6 | **0.025** | 34.0 | 0.964 |
> | Ours ($N=200$) | 0.39 | 95.6 | **0.025** | 33.9 | 0.964 |
> | Ours ($\rho=0.5, E=100$) | **0.38** | 95.7 | 0.028 | 33.9 | 0.964 |
> | Ours ($\rho=2.0, E=100$) | 0.40 | 95.4 | 0.028 | 33.7 | 0.964 |
> | Ours ($\rho=1.0, E=50$) | 0.39 | 95.5 | 0.028 | 33.7 | 0.964 |
> | Ours ($\rho=1.0, E=200$) | **0.38** | 95.7 | 0.028 | 34.0 | **0.965** |
> | Ours ($\rho=1.7, E=1445$) | **0.38** | **95.8** | **0.025** | 33.9 | 0.964 |
> | Ours ($\times 0.25$ triangles) | 0.39 | 95.5 | **0.025** | 34.0 | **0.965** |
> | PhysAvatar [61] | 0.45 | 94.8 | 0.034 | 32.0 | 0.962 |
>
> > **Weakness 3.** "Lack of failure case analysis: The paper primarily focuses on showcasing positive results but does not include any failure cases or challenging scenarios where the method might break down. For instance: (1) Are there cases where the simulation fails due to numerical instabilities or incorrect body-garment collisions? (2) Are certain garment types (e.g., multilayered skirts, long coats) poorly handled?"
>
> > **Limitations 1.** "Limited discussion of limitations: While the authors briefly mention in the conclusion that they do not support relighting or strand-based simulation, a more thorough analysis of the method's limitations is missing."
>
> **Reply:** We thank the reviewer for pointing out the importance of discussing additional failure cases and limitations. Below, we respond to the questions raised:
>
> **(1) “Are there cases where the simulation fails due to numerical instabilities or incorrect body-garment collisions?”**
>
> Throughout the experiments, we did not observe numerical instabilities, as reflected in Tab. 2, where we report a 100% simulation success rate. Indeed, mitigating the numerical instabilities common in existing physics-based avatars with iterative solvers [24, 31] was one of our main goals, which motivated our design of “an MPM-based simulator capable of simulating objects under complex contacts without failure cases via feedforward velocity projection” (L49–51). We also did not noticeably observe incorrect body-garment collisions in our main experiments; however, we discuss other potential limitations in the following paragraph.
>
> **(2) Are certain garment types (e.g., multilayered skirts, long coats) poorly handled?**
>
> We believe that our current **appearance** modeling may be limited in multilayered garment settings, especially for inner garments. Similar to existing optimization-based avatar creation methods [19, 27, 33, 37, 62], we directly optimize the appearance parameters from multi-view RGB videos. As a result, artifacts may occur in regions that were occluded during training (e.g., inner garments occluded by outer garments) but become visible in novel poses — stemming from incomplete supervision for occluded areas. More recent avatar creation methods utilize generative priors to inpaint these occluded regions ([R3]), which can be an important future direction for our work. We will discuss this limitation and potential future work in the revision.
>
>
> [R3] Kwon *et al.*, Generalizable Human Gaussians for Sparse View Synthesis. In ECCV, 2024.

---

> ### Comment · Reviewer_gjLz · 2025-08-05
>
> After the rebuttal, most of my concerns are resolved, thus I will keep my positive score.

---

> > ### Author Response · Authors · 2025-08-05
> >
> > Thank you for your response and for your thoughtful feedback throughout the review process. We're glad to hear that our rebuttal addressed your concerns.

---

### Official Review · Reviewer_1QGS · 2025-07-01

**Clarity:** 3
**Significance:** 3
**Originality:** 3
**Rating:** 4
**Confidence:** 3

**Summary:**

The paper introduces MPMAvatar, a novel framework that constructs 3D clothed human avatars from multi-view videos with robust animation and high-fidelity rendering. The method integrates a tailored Material Point Method (MPM) simulator that includes an anisotropic constitutive model for garment dynamics and a mesh-based collision handling algorithm for efficient and robust simulation. The method demonstrates strong performance over baselines and enables zero-shot generalization to novel interactions.

**Questions:**

1.	It is unclear how the method distinguishes between the garment and the underlying body regions in the avatar mesh. Could the authors clarify the process used to segment or identify these regions?
2.	In my understanding, the rightmost image of Fig. 2 depicts the body grid with blue arrows, the garment grid with green arrows, and the collision regions between them with red arrows. However, I am not entirely certain this interpretation is correct. It would be helpful if the authors could provide a more detailed explanation to clarify the figure and improve overall readability.
3.	Why is the formula for computing the final velocity in the rightmost image of Fig. 2 defined in that specific way? Is there a theoretical foundation behind this formulation?
4.	Are the physical parameter learning and appearance learning processes conducted separately or jointly?

**Ethical Concerns:**

["NO or VERY MINOR ethics concerns only"]

**Final Justification:**

I maintain my positive rating.

**Limitations:**

yes

**Paper Formatting Concerns:**

nil

**Quality:**

3

**Strengths And Weaknesses:**

Strengths:
1.	Tailoring of MPM for effective garment dynamics modeling based on anisotropic constitutive model and mesh-based collider handling is novel.
2.	Both quantitative and qualitative results show clear improvements in dynamics accuracy, rendering fidelity, and simulation robustness and efficiency.
3.	The method successfully handles unseen physical interactions, demonstrating its significant zero-shot generalizable ability.

Weaknesses:
1.	Unlike PhysAvatar, MPMAvatar does not support relighting, which could be critical for some applications.
2.	Some technical descriptions could benefit from clearer exposition or additional illustrations, as certain implementation details may be difficult to follow for readers unfamiliar with MPM or anisotropic constitutive models.
3.	As shown in the supplemental video at 16s, minor holes are visible in the clothing.
4.	The method relies on multi-view video input, which may limit its practicality in real-world applications.
5.	The dynamics of non-garment regions are modeled using linear blend skinning, which may not sufficiently capture realistic motion for complex structures such as hair.

---

> ### Author Rebuttal · Authors · 2025-07-30
>
> # General Response
>
> We thank the reviewers for their constructive feedback and recognizing the **novelty of our collision handling method** (gjLz, 1QGS), **strong quantitative and qualitative results in dynamics, rendering quality, and efficiency** (xXua, gjLz, 1QGS, 1m12), as well as our method’s **zero-shot generalizability to unseen physical interactions** (xXua, 1QGS).
>
> In response to the reviewers’ feedback, we have made several additions to strengthen the paper. Specifically, we conducted three new experiments:
> * **penetration depth analysis** for evaluating contact handling,
> * a **VLM-based assessment** of physical plausibility using PhyGenBench [R1], and
> * **additional ablations** on key simulation hyperparameters.
>
> We also updated Tab. 1 to include comparisons with two recent state-of-the-art baselines [43, R2]. We hope these revisions address the reviewers’ concerns and welcome any further suggestions.
>
> **Table R1:** Quantitative comparison results including new baselines
> | Method | CD ($\times 10^{3}$) (↓) | F-Score (↑) | LPIPS (↓) | PSNR (↑) | SSIM (↑) |
> |----------|----------|----------|----------|----------|----------|
> | MMLPHuman [R2] | 0.47 | 94.9 | 0.039 | 29.3 | 0.954 |
> | Gaussian Garments [43] | 2.39 | 86.5 | 0.042 | 29.5 | 0.959 |
> | PhysAvatar [61] | 0.55 | 92.9 | 0.035 | 30.2 | 0.957 |
> | Ours | **0.42** | **95.7** | **0.033** | **32.0** | **0.963** |
>
> [R1] Meng *et al.* Towards World Simulator: Crafting Physical Commonsense-Based Benchmark for Video Generation, In ICML, 2025.
>
> [R2] Zhan *et al.* Real-time High-fidelity Gaussian Human Avatars with Position-based Interpolation of Spatially Distributed MLPs. In CVPR, 2025.
>
> ---
>
> # Response to Reviewer 1QGS
>
> > **Weakness 1.** "Unlike PhysAvatar, MPMAvatar does not support relighting, which could be critical for some applications."
>
> **Reply:** While we achieve rendering metrics superior to the existing SotA physics-based avatar [61] even without relighting (Tab. 1), we agree that relighting is important for further enhancing the visual realism of our results — as also discussed in L372–373. Following recent relightable Gaussian avatar frameworks [R3, R4], we believe our framework can be extended to support relighting in the future. We will highlight this discussion in the revision.
>
> [R3] Jiang *et al.*, GaussianShader: 3D Gaussian Splatting with Shading Functions for Reflective Surfaces. In CVPR, 2024.
>
> [R4] Saito *et al.*, Relightable Gaussian Codec Avatars. In CVPR, 2024.
>
> > **Weakness 2.** "Some technical descriptions could benefit from clearer exposition or additional illustrations, as certain implementation details may be difficult to follow for readers unfamiliar with MPM or anisotropic constitutive models."
>
> **Reply:** We thank the reviewer for the helpful comment. We acknowledge that some readers may find it difficult to follow the technical details related to MPM [17] and the anisotropic constitutive model [16], especially without prior familiarity. In the main paper, we prioritized clarity in presenting our novel contributions while referring to established literature for existing components due to space limitations. As we agree that additional illustrations and explanations could further improve readability for a broader range of readers, we will include extended technical details and visualizations in the supplementary, following your suggestion.
>
> > **Weakness 3.** "As shown in the supplemental video at 16s, minor holes are visible in the clothing."
>
> **Reply:** Although these minor holes are rare and our method still achieves superior visual results compared to the existing SotA work (PhysAvatar [61]), we believe they are caused by Gaussian splats that were occluded in the training videos but become exposed in the test sequence. We will discuss this in the revision and consider applying generative priors to inpaint these training-time occluded regions (similar to [R5]) as future work.
>
> [R5] Kwon *et al.*, Generalizable Human Gaussians for Sparse View Synthesis. In ECCV, 2024.
>
> > **Weakness 4.** "The method relies on multi-view video input, which may limit its practicality in real-world applications."
>
> **Reply:** Thank you for your comment. Although our multi-view setup mainly follows the existing SotA physics-based avatar work [61], we agree that this input modality limits further practicability in real-world applications. We will include this discussion and also consider supporting a single-view setup as future work — e.g., by using diffusion-based view synthesis to obtain pseudo multi-view supervision from monocular video inputs, similar to a very recent work ([R6]).
>
> [R6] Tang *et al.*, GAF: Gaussian Avatar Reconstruction from Monocular Videos via Multi-view Diffusion. In CVPR, 2025.
>
> > **Weakness 5.** "The dynamics of non-garment regions are modeled using linear blend skinning, which may not sufficiently capture realistic motion for complex structures such as hair."
>
> **Reply:** As also discussed in L373–375, we agree that linear blend skinning for non-garment regions is limited in modeling complex dynamics, especially for hair motion. Since our underlying MPM framework with the anisotropic constitutive model [16] is versatile and supports the modeling of strand-like materials as well, we will consider extending our framework to strand-based hair simulation.
>
> > **Questions 1.** "It is unclear how the method distinguishes between the garment and the underlying body regions in the avatar mesh. Could the authors clarify the process used to segment or identify these regions?"
>
> **Reply:** For garment and human part segmentation, we mainly follow the pre-processing stage used in the existing SotA physics-based avatar work (PhysAvatar [61]). To ensure a fair comparison with [61], we directly use the garment-body segmentation provided by [61], which is manually annotated into garment and body regions. For the 4D-DRESS [51] dataset, we similarly define garment and body regions based on a consistent mesh topology through a manual process. While this segmentation step is currently manual, we believe it can be automated using existing segmentation approaches, as demonstrated in recent avatar methods such as D3GA [62], where per-image masks are projected onto the 3D mesh.
>
> > **Questions 2.** "In my understanding, the rightmost image of Fig. 2 depicts the body grid with blue arrows, the garment grid with green arrows, and the collision regions between them with red arrows. However, I am not entirely certain this interpretation is correct. It would be helpful if the authors could provide a more detailed explanation to clarify the figure and improve overall readability."
>
> **Reply:** Your understanding of Fig. 2 is correct — blue arrows indicate body grid velocities, green arrows denote garment grid velocities, and red arrows show colliding grid regions where velocity projection is applied. Following your comment, we will clarify these arrow types in the figure caption in the revision.
>
> > **Questions 3.** "Why is the formula for computing the final velocity in the rightmost image of Fig. 2 defined in that specific way? Is there a theoretical foundation behind this formulation?"
>
> **Reply:** The overall procedure for computing the final velocity in Fig. 2 is mainly adopted from standard practices used in MPM [17]. In particular, projecting out the inward normal component of the relative velocity between the garment and the collider is a conventional strategy for contact handling in continuum mechanics. Our specific contribution lies in making this velocity projection feasible for complex colliders represented as meshes, where collider velocity and normals are strictly defined on the discretized surface, not on the grid nodes — unlike simple, analytic colliders typically previously considered in MPM [17]. Following your comment, we will explain this point more clearly.
>
> > **Questions 4.** "Are the physical parameter learning and appearance learning processes conducted separately or jointly?"
>
> **Reply:** We clarify that physical parameter learning and appearance learning are conducted separately, similar to the existing SotA physics-based work (PhysAvatar [61]). Following your comment, we will describe this more clearly in the revision.

---

> > ### Comment · Reviewer_1QGS · 2025-08-04
> >
> > Thanks for the response. My questions have been well addressed.

---

> > > ### Author Response · Authors · 2025-08-04
> > >
> > > Thank you for your response, and once again, for your constructive reviews, which have helped improve our manuscript.
> > >
> > > We’re glad to hear that our rebuttal has addressed your questions. If your concerns have been sufficiently resolved, we would greatly appreciate it if you could kindly consider updating your rating.
> > >
> > > Thank you again for engaging in the discussion with us.

---

### Official Review · Reviewer_1m12 · 2025-07-04

**Clarity:** 2
**Significance:** 2
**Originality:** 2
**Rating:** 4
**Confidence:** 5

**Summary:**

The paper presents MPM-Avatar, a method for learning 3D avatars with physically grounded deformation behavior using   Material Point Method (MPM) simulations. The approach leverages sparse 3D Gaussian representations to model both geometry and appearance, while also incorporating MPM-based losses to ensure plausible physical responses to external forces. The system jointly optimizes visual fidelity and dynamic realism from monocular RGB videos and physical annotations. Key contributions include the integration of physics priors into Gaussian-based avatar learning, a differentiable MPM framework for supervision, and experiments demonstrating improved physical plausibility over prior purely visual methods.

**Questions:**

1. Cite relevant prior work and clarify distinctions: Please cite key missing works in differentiable cloth and avatar simulation, including Differentiable Cloth Simulation for Inverse Problems (Liang et al., NeurIPS 2019), DiffCloth (Li et al., SIGGRAPH 2022), and DiffAvatar (Li et al., CVPR 2024). These works are directly relevant to your goals and share conceptual ground in learning physically realistic avatars. In your revision, please clearly articulate how your approach differs from and improves upon these methods, especially in terms of simulation modeling (e.g., MPM vs. mass-spring/cloth) and supervision (e.g., RGB + physics vs. mesh-based optimization).

2. Evaluation of contact handling: The paper introduces contact-aware physics supervision, but there is no targeted quantitative evaluation of this component. Could the authors include an ablation or metric-based analysis (e.g., contact force accuracy, penetration depth, collision fidelity) to isolate and verify the effectiveness of the contact modeling? This would help clarify how much contact dynamics contribute to the observed improvements.

3. Quantitative evaluation of physical plausibility: While qualitative examples are strong, can you provide more quantitative metrics to assess physical realism? This would help substantiate the claim of improved physical behavior.

4. Scalability and computational cost: What is the training and inference cost of incorporating MPM simulation-based gradients, especially using finite differences? Could you provide a breakdown of runtimes on the over learning process and each iteration (forward pass, MPM loss computation, Gaussian updates)? This is important for judging the method’s practicality and applicability in larger-scale setups.

**Ethical Concerns:**

["NO or VERY MINOR ethics concerns only"]

**Final Justification:**

The authors' response has clarified my questions so I'm raising my score to borderline accept.

**Limitations:**

The authors do discuss limitations, particularly regarding physical supervision and force annotations. However, one important limitation not mentioned is the scalability of the framework with respect to the number of physical parameters. The current optimization procedure resembles that of PhysAvatar, which used a non-differentiable simulator and required two simulations per parameter, resulting in linear runtime scaling. If the same approach is used here, the method may become computationally infeasible for identifying a large number of physical parameters. Clarifying this limitation and discussing possible mitigation strategies (e.g., using differentiable simulators) would strengthen the discussion.

**Quality:**

3

**Strengths And Weaknesses:**

The paper proposes a framework that integrates Material Point Method (MPM) simulations with 3D Gaussian splatting for learning avatars that exhibit physically grounded deformations from monocular video. Experiments show quantitative and  qualitative improvements compared to PhysAvatar.

Regarding related works, there is a lack of discussion and citation of several foundational works in differentiable cloth simulation that are directly relevant to the avatar reconstruction problem. In particular, Differentiable Cloth Simulation for Inverse Problems (Liang et al., NeurIPS 2019), DiffCloth (Li et al., SIGGRAPH 2022), and DiffAvatar (Li et al., CVPR 2024) present physically grounded, differentiable simulation frameworks that have been applied to cloth and digital avatars and addressing key aspects such as collisions.  These works should be cited and contrasted to clarify the novelty and scope of the current method.

 In addition, while the paper introduces a contact-aware physics loss via MPM simulation, it lacks a dedicated quantitative evaluation of the contact handling. There is no ablation or metric provided to demonstrate how well the method models contact dynamics compared to baselines or ground truth, which limits the ability to assess this component’s contribution.

---

> ### Author Rebuttal · Authors · 2025-07-30
>
> # General Response
>
> We thank the reviewers for their constructive feedback and recognizing the **novelty of our collision handling method** (gjLz, 1QGS), **strong quantitative and qualitative results in dynamics, rendering quality, and efficiency** (xXua, gjLz, 1QGS, 1m12), as well as our method’s **zero-shot generalizability to unseen physical interactions** (xXua, 1QGS).
>
> In response to the reviewers’ feedback, we have made several additions to strengthen the paper. Specifically, we conducted three new experiments:
> * **penetration depth analysis** for evaluating contact handling,
> * a **VLM-based assessment** of physical plausibility using PhyGenBench [R1], and
> * **additional ablations** on key simulation hyperparameters.
>
> We also updated Tab. 1 to include comparisons with two recent state-of-the-art baselines [43, R2]. We hope these revisions address the reviewers’ concerns and welcome any further suggestions.
>
> **Table R1:** Quantitative comparison results including new baselines
> | Method | CD ($\times 10^{3}$) (↓) | F-Score (↑) | LPIPS (↓) | PSNR (↑) | SSIM (↑) |
> |----------|----------|----------|----------|----------|----------|
> | MMLPHuman [R2] | 0.47 | 94.9 | 0.039 | 29.3 | 0.954 |
> | Gaussian Garments [43] | 2.39 | 86.5 | 0.042 | 29.5 | 0.959 |
> | PhysAvatar [61] | 0.55 | 92.9 | 0.035 | 30.2 | 0.957 |
> | Ours | **0.42** | **95.7** | **0.033** | **32.0** | **0.963** |
>
> [R1] Meng *et al.* Towards World Simulator: Crafting Physical Commonsense-Based Benchmark for Video Generation, In ICML, 2025.
>
> [R2] Zhan *et al.* Real-time High-fidelity Gaussian Human Avatars with Position-based Interpolation of Spatially Distributed MLPs. In CVPR, 2025.
>
> ---
>
> # Response to Reviewer 1m12
>
> > **Weakness 1.** "There is a lack of discussion and citation of several foundational works in differentiable cloth simulation that are directly relevant to the avatar reconstruction problem."
>
> > **Questions 1.** "Cite relevant prior work and clarify distinctions."
>
> **Reply:** Thank you for suggesting these additional related works. While our main paper focused on discussing and experimentally comparing against all baselines presented in the most recent SotA work directly addressing the same problem as ours (PhysAvatar [61]), we agree that the suggested works are also valuable in the broader context. We will make sure to include a discussion of them in the revision. In a nutshell:
>
> **DiffAvatar** [R5] is a framework designed to reconstruct a physics-based garment **from a static 3D scan, without modeling appearance**. We think its problem scope differs from ours, which targets photorealistic dynamic avatar reconstruction **from multi-view RGB videos**, with some of our key technical contributions include: (1) **photorealistic appearance modeling** based on Gaussian Splatting with quasi-shadowing (achieving SotA appearance results in Tab. 1), and (2) **learning the dynamic behavior of garments** from real observations (achieving SotA dynamics modeling accuracy in Tab. 1).
>
> **Liang** ***et al.*** [R3] and **DiffCloth** [R4] are related to ours in that they propose differentiable cloth simulators. However, as PhysAvatar [61] already noted, [R4] is difficult to “be directly applied due to limitations in modeling complex body colliders.” Indeed, we find that both [R3, R4] primarily consider simulation settings without any colliders, or only with simple geometric colliders (e.g., spheres). In contrast, our target problem of clothed avatar reconstruction requires handling collisions between garments and complex external colliders (e.g., body meshes), which was also a key aspect of our technical contribution. Accordingly, we prioritized discussing methods that address the same clothed avatar reconstruction task using simulators specialized for handling complex collisions (PhysAvatar [61] based on C-IPC [24]).
>
> Nevertheless, we believe that discussing these related works adds valuable context, and we will revise our draft accordingly. Thank you again for your suggestions.
>
> [R3] Liang *et al.* Differentiable Cloth Simulation for Inverse Problems. In NeurIPS, 2019.
>
> [R4] Li *et al.* DiffCloth: Differentiable Cloth Simulation with Dry Frictional Contact. In SIGGRAPH, 2022.
>
> [R5] Li *et al.* DiffAvatar: Simulation-Ready Garment Optimization with Differentiable Simulation. In CVPR, 2024.
>
> > **Weakness 2.** "It lacks a dedicated quantitative evaluation of the contact handling."
>
> > **Questions 2.** "Evaluation of contact handling"
>
> **Reply:** Thank you for your suggestion. However, we would like to clarify that, in our targeted problem – learning physics-based avatars from **real multi-view RGB video observations** – the existing datasets do not provide the ground truth contact force and collision labels, making it non-trivial to measure the suggested contact force accuracy and collision fidelity metrics.
>
> While existing published works addressing the same problem as ours (PhysAvatar [61], Dressing Avatars [53]) and even those proposing novel contact solvers (IPC [23,24]) similarly do not report these two metrics for this reason, they evaluate performance by measuring the discrepancy between the ground-truth geometry (which is obtainable from existing datasets [14, 51]) and the simulated geometry, which is a byproduct of the proposed contact modeling, thus somewhat **indirectly** reflecting the contact modeling accuracy. Regarding these geometry-based metrics, we reported that our method outperforms the existing SotA [61] in Tab. 1 in the paper.
>
> Nevertheless, the suggested penetration depth metric is still measurable without access to ground-truth contact assets, and we present additional experimental results in Tab. R2 below. We find that our method yields lower penetration depth than the existing SotA method [61] on the ActorsHQ dataset, achieving over a 6× reduction in the average value. Following your suggestion, we will include these experimental results in the revision.
>
> **Table R2**
> | Method | Penetration Depth (mm) (↓) |
> |----------|----------|
> | **Ours** | **0.047** |
> | **PhysAvatar** [61] | **0.294** |
>
> > **Questions 3.** "Quantitative evaluation of physical plausibility"
>
> **Reply:** We appreciate the reviewer’s suggestion. We agree that physical plausibility is an important property to evaluate; however, we would like to note that there is no standard metric for quantitatively measuring physical plausibility in our related works [53, 61]. If the reviewer is referring to a specific metric in this context, we would appreciate it if they could kindly clarify, and we would be happy to discuss it further.
>
> Nevertheless, we additionally conducted experiments using two metrics that we believe reflect physical plausibility. For the first metric, we considered penetration depth (also mentioned by the reviewer in another comment), as shown above in Tab. R2, where our method outperforms the baseline (PhysAvatar [61]).
>
> For the second metric, we adopt the Key Physical Phenomena Detection metric proposed in PhyGenBench [R1], which computes a plausibility score using a VLM based on how well a given video aligns with physical plausibility prompts. In Tab. R3, we show that our method achieves a plausibility score closer to the ground-truth upper bound than PhysAvatar [61], indicating better alignment with the physical plausibility prompts.
>
> **Table R3**
> | Method | Score (↑) |
> |----------|----------|
> | **GT** | **1.86** |
> | **Ours** | **1.83** |
> | **PhysAvatar [61]** | **1.78** |
>
> > **Questions 4.** "Scalability and computational cost"
>
> > **Limitations 1.** "scalability of the framework with respect to the number of physical parameters."
>
> **Reply:** Thank you for your insightful feedback. As mentioned, our method uses a finite-difference approach, which requires $n+1$ forward simulations to optimize $n$ physical parameters, resulting in linear runtime scaling with respect to the number of parameters. While this can indeed be “computationally infeasible for identifying a large number of physical parameters,” as pointed out in the comment, we would like to note that in our specific target application—physics-based dynamic avatar reconstruction—the garment behavior is mostly homogeneous. Thus, a single-material assumption **per garment** (which is also adopted in prior work [61]) was sufficient to achieve new SotA dynamics accuracy, without requiring the identification of a larger number of physical parameters.
>
> Nevertheless, we agree that scalability is an important consideration for broader future applications. We will discuss this limitation and possible mitigation strategies (e.g., differentiable simulators) in the revision.
>
> Regarding the training and inference costs mentioned in the comment:
> * Our training takes approximately **5–8 hours per garment**, which is substantially faster than the existing SotA method targeting the same problem (PhysAvatar [61]), which reports **1–2 days** of training time.
> * Our inference time is **1.1s per frame**, as reported in Tab. 2 of the paper, representing a **~155× speed-up** compared to [61].
> * Below is a breakdown of the runtimes:
>   * **Simulation forward pass**: 1.1s per frame on average (Tab. 2).
>   * **MPM loss computation**: Requires $n+1=4$ forward passes per step to optimize $n=3$ physical parameters, resulting in approximately $4 \times 24 \times 1.1$s $\approx \mathbf{100}$**s** per iteration. This linear scaling behavior is also shared by PhysAvatar [61], which adopts the same finite-difference scheme.
>   * **Gaussian update**: Takes **~0.04s per iteration** via back propagation and adaptive density control; total time for optimizing Gaussian Splat parameters is **~1.5 hours per subject** (for 30K steps). Note that this stage is **independent of the number of physical parameters.**

---

### Note · Authors · 2025-08-14

We sincerely thank the AC and all reviewers for their time, thoughtful feedback, and constructive discussions. The reviewers’ insights have been invaluable in improving our manuscript, and we deeply appreciate the effort and attention devoted to the review process.

Following the rebuttal, we are glad that all reviewers acknowledged their concerns and questions had been well addressed. To address the *initial* concerns regarding additional evaluations on physical plausibility, collision-handling accuracy, and hyperparameter sensitivity, we presented further experimental analyses in the rebuttal, including:

- **Penetration depth analysis** to evaluate contact handling,
- **VLM-based assessment** of physical plausibility, and
- **Additional ablations** on key simulation hyperparameters,

along with comparisons to recent SotA baselines published in parallel with our submission [R1, 43].

Through the review process, we were particularly encouraged that all reviewers expressed positive views on our work and recognized its key strengths, including:

- **Substantial performance improvements over SotA [61] (1m12, 1QGS, gjLz, xXua) across multiple aspects:**

    (a) *Dynamics modeling accuracy (1QGS)*: Over 24% improvement in Chamfer distance compared to [61].

    (b) *Rendering accuracy (xXua)*: Superior rendering via 3D Gaussian Splatting with quasi-shadowing (vs. mesh-based rendering in [61]).

    (c) *Robustness (1QGS, xXua)*: 100% simulation success rate (vs. 37.6% for [61]), enabled by our novel MPM-based simulator.

    (d) *Efficiency (gjLz, xXua)*: Over 150× faster simulation than [61].

    (e) *Zero-shot generalizability to interactions in unseen scenes (1QGS, xXua)*: To the best of our knowledge, the first avatar framework to demonstrate this capability.

- **Technically sound and novel method (1QGS, gjLz)**: Tailored MPM with anisotropic modeling and mesh-based collision handling.
- **Comprehensive ablation study (xXua)**: Detailed analyses isolating the contribution of each proposed component.

We sincerely appreciate the reviewers’ positive comments and will ensure that we incorporate all additional suggestions discussed during the rebuttal. Once again, we thank the AC and reviewers for helping us improve our manuscript.

[R1] Zhan *et al.* Real-time High-fidelity Gaussian Human Avatars with Position-based Interpolation of Spatially Distributed MLPs. In CVPR, 2025.

---

### Decision · Program_Chairs · 2025-09-17

**Decision:**

Accept (poster)

**Comment:**

Concerns raised during the review process were addressed in the rebuttal with additional experiments and explanations. All reviewers maintained or increased their positive scores. For the camera-ready version, under the pape limit, the authors should: (1) add missing citations and comparison with relevant differentiable simulation work, (2) expand quantitative evaluation of contact handling and physical plausibility, (3) improve clarity of figures and technical exposition (e.g., velocity formulations, edge vectors), and (4) provide a clearer discussion of limitations and assumptions.